# Cyclic peptide FXII inhibitor provides safe anticoagulation in a thrombosis model and in artificial lungs

Jonas Wilbs[1], Xu-Dong Kong [1], Simon J. Middendorp[1], Raja Prince[2,3], Alida Cooke[4], Caitlin T. Demarest[4], Mai M. Abdelhafez [3], Kalliope Roberts [4], Nao Umei [4], Patrick Gonschorek [1], Christina Lamers [1], Kaycie Deyle[1], Robert Rieben [3], Keith E. Cook [4], Anne Angelillo-Scherrer[2,3] & Christian Heinis [1✉]

Inhibiting thrombosis without generating bleeding risks is a major challenge in medicine. A promising solution may be the inhibition of coagulation factor XII (FXII), because its knock-out or inhibition in animals reduced thrombosis without causing abnormal bleeding. Herein, we have engineered a macrocyclic peptide inhibitor of activated FXII (FXIIa) with sub-nanomolar activity ($K_i = 370 \pm 40$ pM) and a high stability ($t_{1/2} > 5$ days in plasma), allowing for the preclinical evaluation of a first synthetic FXIIa inhibitor. This 1899 Da molecule, termed FXII900, efficiently blocks FXIIa in mice, rabbits, and pigs. We found that it reduces ferric-chloride-induced experimental thrombosis in mice and suppresses blood coagulation in an extracorporeal membrane oxygenation (ECMO) setting in rabbits, all without increasing the bleeding risk. This shows that FXIIa activity is controllable in vivo with a synthetic inhibitor, and that the inhibitor FXII900 is a promising candidate for safe thromboprotection in acute medical conditions.

[1] Institute of Chemical Sciences and Engineering, Ecole Polytechnique Fédérale de Lausanne (EPFL), CH-1015 Lausanne, Switzerland. [2] Department of Hematology and Central Hematology Laboratory, Inselspital, Bern University Hospital, University of Bern, CH-3010 Bern, Switzerland. [3] Department of Clinical Research, University of Bern, CH-3008 Bern, Switzerland. [4] Department of Biomedical Engineering, Carnegie Mellon University, Pittsburgh, PA 15213, USA. ✉email: christian.heinis@epfl.ch

Coagulation stops the bleeding at sites of vessel wall injuries, but excess coagulation is equally dangerous as it can lead to thrombosis. Blood vessel blockages can occur in the arterial and venous circulation and cause myocardial infarction, ischemic stroke, and pulmonary embolism, which collectively are the leading causes of disability and death in the industrialized world[1]. In all of the anticoagulants widely used for the acute and prophylactic treatment of thrombosis[2,3], bleeding is a common side effect, and the initiation of any therapy to treat thrombotic disorders must always weigh these risks and benefits, considering the severity of the potential side effects. Thus, there is a growing need for the development of effective anticoagulants that ideally would not impair hemostatic ability[4].

A promising novel perspective for developing safe anticoagulants with reduced or no bleeding risks is the inhibition of coagulation factors XII (FXII) and XI (FXI), both proteases of the intrinsic coagulation pathway[5,6]. FXII is the initiating protease of the procoagulant and proinflammatory contact system, and it drives both the intrinsic pathway of coagulation and the kallikrein-kinin system[7]. Mice lacking FXII display reduced thrombosis in mouse models of injury-induced arterial and venous thrombosis[8,9] and are protected from pathological thrombosis in cerebral ischemia[10]. At the same time, humans naturally lacking FXII and FXII-knockout mice have a normal hemostatic capacity and do not bleed abnormally[8,11], which supports the idea that drugs targeting the protease could potentially display antithrombotic effects without significantly compromising hemostasis. Concordantly, the reduction of FXII expression by antisense oligonucleotides suppresses thrombosis in arterial and venous thrombosis mouse models and catheter thrombosis in rabbits[12,13]. The inhibition of FXII by protein-based inhibitors, such as antibodies[14,15] or insect- and plant-derived proteins[16–18], also reduces thrombosis in mouse, rat, rabbit, and primate models of induced arterial or venous thrombosis and shows potential avenues for therapeutic anticoagulation.

FXII-driven blood coagulation is a major challenge in cardiopulmonary bypass (CPB) surgeries in which a heart-lung machine temporarily supports the circulation. Contact between FXII and artificial surfaces, such as the oxygenator membrane or tubing, induces a conformational change that leads to the proteolytic activation of the FXII zymogen that turns on the contact system. Activation of the procoagulant intrinsic coagulation pathway and the proinflammatory kallikrein-kinin system leads to blood clotting and inflammation[19]. Contact activation also poses a problem in extracorporeal membrane oxygenation (ECMO), a medical procedure in which an artificial lung system is used for longer periods of time as a life support for patients with severe cardiac and/or pulmonary failure[20]. The standard strategy for suppressing contact system activation in extracorporeal circuits relies on high doses of heparin, which also inhibit proteases of the extrinsic and common coagulation pathway and thus bear an inherent risk of bleeding[21]. Several strategies for suppressing contact activation were established in experimental models, but heparin remains the anticoagulation strategy of choice[22]. In a recent study, Renné and co-workers showed that a human FXIIa-inhibiting antibody, 3F7 ($IC_{50} = 13$ nM), prevents clotting and thrombosis in a cardiopulmonary bypass system in rabbits without increasing therapy-associated bleeding[14], indicating the usefulness of targeting FXII.

FXII is implicated in several other medical conditions[23], including hereditary angioedema (HAE)[24], reperfusion injury[25], Alzheimer's disease[26,27], and multiple sclerosis[28], meaning that potential FXIIa inhibitors could be used in a variety of treatments not limited to thrombosis prevention. In Type III HAE, for example, a mutation in FXII facilitates its activation, leading to

excessive bradykinin release that causes edema[29]. An improved variant of the above-mentioned FXIIa inhibitory antibody 3F7 has entered a phase I clinical evaluation for the treatment of HAE[30].

While high-affinity protein-based FXIIa inhibitors were successfully generated over the last years[31], the development of synthetic, small-molecule inhibitors has been more challenging. Small molecules have a number of strengths that make them attractive for drug development, including a uniform composition, efficient tissue penetration, high stability, low immunoreactivity, and ease of production by chemical synthesis. The best small molecule FXIIa inhibitors reported to date are the coumarin derivative 44 ($IC_{50} = 4.4$ µM)[32] and H-D-Pro-Phe-Arg-chloromethylketone (PCK; $IC_{50} = 0.18$ µM)[10]. While they have proven to be useful as research compounds in FXIIa inhibition studies, their covalent inhibition mechanism and their moderate potency and selectivity limit their drug development potential.

We recently identified high-affinity FXIIa inhibitors based on cyclic peptides using phage display, including the bicyclic peptide FXII618 ($K_i = 8.1$ nM)[33]. In previous work, the substitution of individual amino acids in FXII618 to unnatural ones further improved the affinity[34,35], though due to their limited binding to animal FXIIa homologs and low proteolytic stability in plasma, these particular inhibitors could not be evaluated in vivo. For this reason, we describe herein the further improvement of the inhibitor to achieve sub-nanomolar potency towards human, mouse, and rabbit FXIIa, and we enhanced the stability in plasma to several days (half-life in human plasma ex vivo). These properties allowed for the pre-clinical evaluation of the inhibitor in two animal models. This work shows that synthetic FXIIa inhibitors can be developed to efficiently prevent thrombosis in mice and suppress coagulation in artificial lungs in rabbits without increasing the risk of bleeding.

## Results

**Engineered macrocycle inhibits FXIIa with sub-nanomolar affinity.** In a previously developed FXII inhibitor, FXII618, we showed that the N-terminal arginine (Arg1) is rapidly cleaved by plasma proteases ($t_{1/2}$ around 4 h in human plasma), causing a 40-fold reduction in the $K_i$ of the inhibitor[34]. When Arg1 was substituted with diverse unnatural amino acids, the stability could be improved at the price of a weaker inhibition, wherein amino acids with positively charged side chains showed the smallest affinity losses[34]. In a new attempt, we replaced Arg1 with a panel of di-peptides that could potentially reach a larger surface area on FXIIa to form productive interactions (Fig. 1a, Supplementary Table 1). Both amino acids in the di-peptides were D-amino acids to prevent proteolysis, and one of the two was D-Arg to maintain a positive charge. To avoid extensive purifications for the initial screening tests, we developed a synthesis and screening approach based on coumarin-labeled peptides to compare the crude activities, as described in detail in the Methods. A stability assay showed that the di-peptides based on D-amino acids were resistant to proteolysis, prolonging the plasma half-life around fivefold to 20 h (Supplementary Fig. 1), and several di-peptides improved the $K_i$ values around two-fold over FXII618 (Fig. 1a). The best inhibitor, Arg1→D-Arg-D-Ser (FXII850) was synthesized without the coumarin label and showed a $K_i$ of 5 nM, which was a 1.6-fold improvement compared to FXII618 ($K_i = 8.1 \pm 0.7$ nM; Fig. 1b and c, Supplementary Fig. 2). We also tested a panel of diverse amino acid substitutions for Arg8, a position in FXII618 that had not been systematically optimized, using the same screening strategy based on coumarin-labeled peptides (Fig. 1a, Supplementary Table 1). The substitution Arg8→His (FXII851) did not affect the $K_i$ of human FXIIa though had a clear

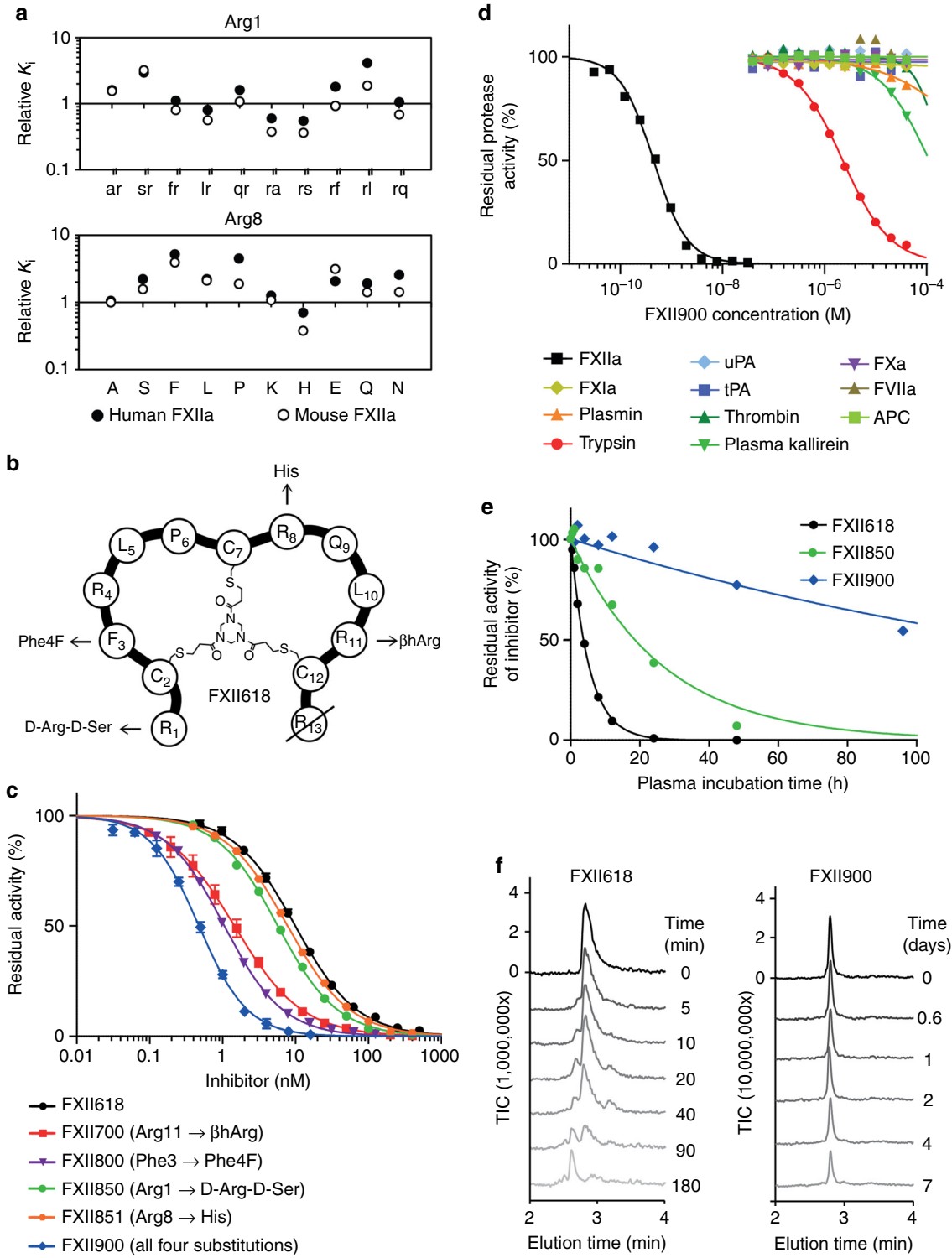

effect on mouse FXIIa inhibition, with $K_i$ values of 6.5 nM and 36 nM, for human and mouse respectively, which corresponded to 1.2- and 2.4-fold improvements over FXII618 (Fig. 1b and c, Supplementary Table 2).

We next combined the beneficial amino acid substitutions identified in this work, the Arg1→D-Arg-D-Ser (FXII850) and Arg8→His (FXII851) with the two previously identified substitutions, Arg11→(S)-$\beta^3$-homoarginine ($\beta$hArg; FXII700; $K_i = 1.5 \pm 0.1$ nM)[35] and Phe3→4-fluorophenylalanine (Phe4F; FXII800; $K_i = 0.84 \pm 0.03$ nM)[34] (Fig. 1b and c, Supplementary Table 2). In

addition, we deleted the C-terminal Arg13 that did not contribute to the binding affinity. The resulting inhibitor, FXII900, blocked human and mouse FXIIa with $K_i$ values of 0.37 ± 0.04 nM and 0.45 ± 0.11 nM, respectively, and was thus substantially more potent than any of its precursors (Fig. 1c). This final inhibitor could be efficiently synthesized in gram-scale by solid-phase peptide synthesis of the linear precursor and subsequent macrocyclization of the crude peptide with the 1,3,5-triacryloyl-1,3,5-triazinane linker (TATA). Only a single HPLC purification step was required to obtain > 95% pure product with an isolated

**Fig. 1 Improving affinity and stability of bicyclic peptide FXIIa inhibitor. a** Amino acids Arg1 and Arg8 of FXII618 were individually substituted to two D-amino acids or natural amino acids, respectively. The relative $K_i$ values for human and mouse FXIIa are indicated as compared to the lead peptide FXII618. Average values of two measurements are shown for peptides with improved affinity. **b** Schematic representation of the lead peptide FXII618. Amino acid substitutions that improve the activity and/or stability are indicated. **c** Inhibition of human FXIIa by FXII618, variants of FXII618 containing a single amino acid substitution, and FXII900 in which four beneficial modifications were combined and Arg13 was deleted. Residual FXIIa activity was determined in two (FXII850, FXII851), three (FXII618, FXII700, FXII800) or five (FXII900) independent measurements. Mean values are indicated. For inhibitors that were tested three or five times, data are presented as mean ± SD. **d** Specificity profiling of FXII900. Residual activities of human FXIIa and ten homologous human proteases were measured. Inhibition of FXII was determined in five, plasmin and APC in three, and all other proteases in two independent measurements. Means values are indicated. **e** Proteolytic stability of FXII900 and two precursors. Bicyclic peptide was incubated in human plasma at 37 °C for the indicated time periods, and the remaining inhibitor activity was quantified in a FXIIa activity assay using fluorogenic substrate. Two independent measurements were performed for FXII850 and FXII900, and four for FXII618. Means values are indicated. **f** Proteolytic stability of FXII618 and FXII900. Bicyclic peptide was incubated in plasma at 37 °C for the indicated time periods and analyzed by LC-MS. Total ion count (TIC) is shown. Source data of **a**, **c**–**e** are provided in a Source data file.

yield greater than 50% (Supplementary Fig. 3). We had previously developed a structural model of the precursor peptide FXII618 bound to FXIIa, in which we could determine with high confidence the positions and interactions of the amino acids Phe3 to Pro6[33]. The newly identified beneficial amino acid substitutions lie outside this region and we could thus not use the model to rationalize the molecular basis of the achieved affinity enhancements.

Finally, a specificity profiling against a panel of homologous plasma proteases was performed to indicate any off-target interactions that would occur in physiological conditions. Activity assays against this chosen panel in the presence of the inhibitor showed that FXII900 is highly selective (Fig. 1d, Supplementary Table 3). All physiologically relevant proteases displayed 100,000-fold or higher selectivity ($K_i$ values > 40 µM), with only trypsin showing any significant inhibition, which occurred at low micromolar concentrations ($K_i = 1.46$ µM).

**FXIIa inhibitor is stable in human plasma for several days**. To measure the stability of the inhibitor in plasma, FXII900 was incubated in human plasma at 37 °C for extended time periods and then the remaining FXIIa inhibitory activity was quantified (Fig. 1e). Mass spectrometric analysis of the various plasma samples showed only FXII900 and no degradation products, though the quantity of the intact inhibitor decreased over time (Fig. 1f, right panel). This suggested that after an initial cleavage event, the inhibitor was rapidly degraded and, therefore, the cleavage products were not detectable. Even so, the half-life of FXII900 was 128 h, around 25-fold longer than that of FXII618. It was also much longer than that of FXII850 carrying only the Arg1→D-Arg-D-Ser modification ($t_{1/2} = 18.5$ h; Fig. 1e), indicating that more than one of the introduced mutations contributed to the improved stability. To identify which of the modifications was most important for stability, we individually reverted the amino acid substitutions (Supplementary Fig. 4 and Supplementary Table 4) and retested the variants, revealing that the Arg11→βhArg substitution contributed the most to the stability improvement (Supplementary Fig. 5).

**FXII900 efficiently inhibits the intrinsic coagulation pathway ex vivo**. To assess the ability of FXII900 to block FXII-driven blood plasma coagulation and to further evaluate its selectivity, we performed coagulation tests to measure activated partial thromboplastin time (aPTT) and prothrombin time (PT). aPTT and PT measure the time until coagulation upon initiation of the intrinsic and extrinsic pathways, respectively, meaning that selective FXIIa inhibition would prolong aPTT but not PT, and results are reported in terms of the concentration of the tested compound required for a 1.5x increase in coagulation time

($EC_{1.5x}$). FXII900 prolonged aPTT in human plasma with an $EC_{1.5x}$ of 0.79 µM, thus 4.3-fold better than its precursor FXII618 ($EC_{1.5x}$ of 3.4 µM; Fig. 2a). FXII900 did not affect PT, even at the highest concentration tested (120 µM), confirming the selectivity for the FXIIa target (Fig. 2b). The inhibitor also prolonged the aPTT in plasma of all other species, namely mouse ($EC_{1.5x} = 2.9$ µM), rabbit ($EC_{1.5x} = 0.102$ µM), and pig ($EC_{1.5x} = 12.2$ µM), wherein the potency varied significantly (Fig. 2a). For rabbit plasma, we observed a strong prolongation of aPTT at low inhibitor concentrations and a sharp transition to a maximal effect. This pattern was previously seen for antibody-based FXIIa inhibitors tested in rabbit plasma[14], and we thus expected that this was a rabbit plasma-specific phenomenon and not based on a particularly strong affinity for rabbit FXIIa. The inhibitor did not prolong PT in any of the three animal plasma tested (Fig. 2b).

To assess if FXII900 prolonged aPTT in the plasma of the four species to different extents due to varying $K_i$s for the various FXII orthologues, we cloned and expressed rabbit and pig FXII because they were not commercially available (Supplementary Figs. 6 and 7) and tested their inhibition by FXII618 and FXII900 (Fig. 2c). These experiments, described in detail in the Supplementary Information, showed that the FXII orthologues were inhibited with different strengths and that the extent of aPTT prolongation correlated with FXII inhibition ($K_i$).

**Pharmacokinetics in mice, rabbits, and pigs**. We assessed the pharmacokinetic properties in three species, mice, rabbits, and pigs, because suitable models for thrombotic diseases are available for these animals. In mice, we determined the pharmacokinetics following subcutaneous administration (5 mg kg$^{-1}$, $n = 3$, different mice for each time point). FXII900 reached a plasma concentration of around 1 µM after 3 min and remained above this concentration for 30 min as determined by LC-MS (Fig. 3a, upper panel). The aPTT was prolonged as expected based on the plasma concentrations; namely, it was extended by two-fold at 15 min and remained prolonged by over 1.5-fold until 30 min (Fig. 3a, lower panel).

In rabbits, we determined the pharmacokinetic properties for intravenous and subcutaneous administration. Upon intravenous administration ($n = 3$, 3.7 mg kg$^{-1}$), FXII900 showed an elimination half-life of $12 ± 2$ min (Fig. 3b, upper panel; pharmacokinetic parameters provided in Supplementary Table 5). The aPTT was initially more than eight-fold prolonged and remained more than three-fold prolonged for the entire time monitored (40 min; Fig. 3b, lower panel). When applying the same dose subcutaneously ($n = 4$, 3.7 mg kg$^{-1}$), the peptide remained in a narrower concentration range, staying above 100 nM and below 300 nM for between 10 and 80 min after administration (Fig. 3c, upper panel). The aPTT with the subcutaneous administration was prolonged by more than 2-fold for 40 min (Fig. 3c, lower panel).

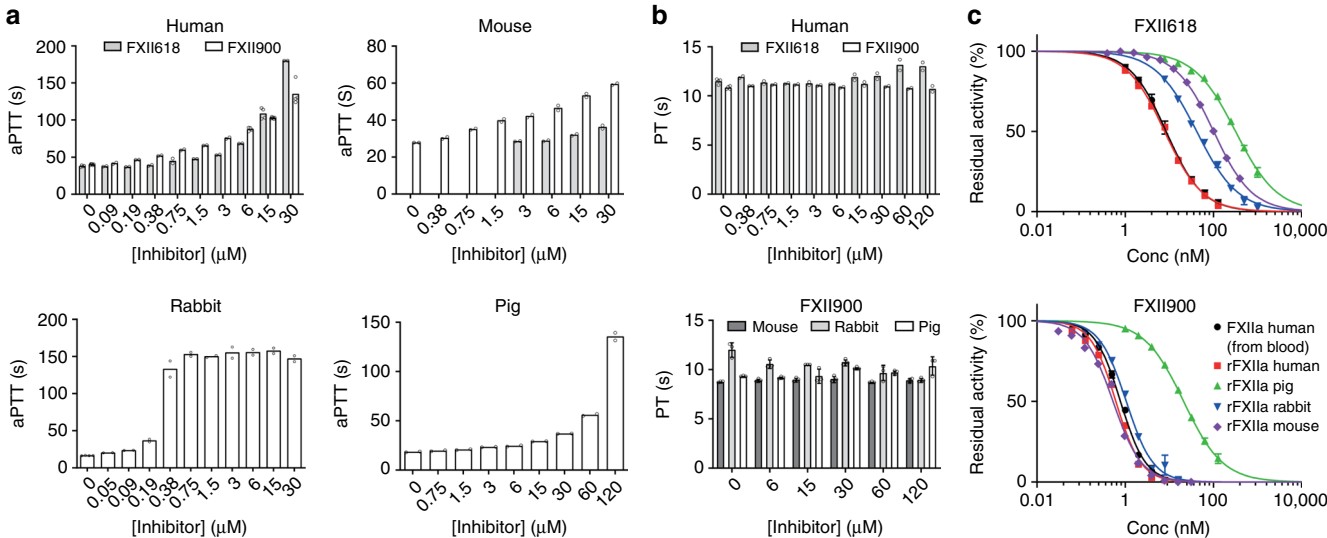

**Fig. 2 Inhibiting FXIIa of different species. a** Inhibition of the intrinsic coagulation pathway in human, mouse, rabbit, and pig plasma ex vivo. The aPTT in the presence of different concentrations of FXII900 is shown. For human and mouse plasma, the aPTT of FXII618 was measured for comparison. All coagulation times were determined in two, four or six independent measurements. Mean values are indicated. Data of individual measurements are shown as dots. **b** PT in the presence of different concentrations of FXII900 and FXII618 in human plasma (upper panel), and FXII900 in mouse, rabbit, and pig plasma (lower panel). PT was determined in two (human plasma) or three independent measurements (all animal plasmas). Mean values are indicated. Data of individual measurements are shown as dots. For the animal plasmas, means ± SD is indicated. **c** Inhibition of recombinant FXIIa (rFXIIa) from different species by FXII618 and FXII900. The inhibition of human FXIIa derived from blood is shown for comparison (data from Fig. 1c). Residual FXIIa activities were determined in two (human, mouse) or three (pig, rabbit) independent measurements. Mean values are indicated. For pig and rabbit FXIIa inhibition, the data are presented as mean ± SD. Source data of **a**–**c** are provided as a Source data file.

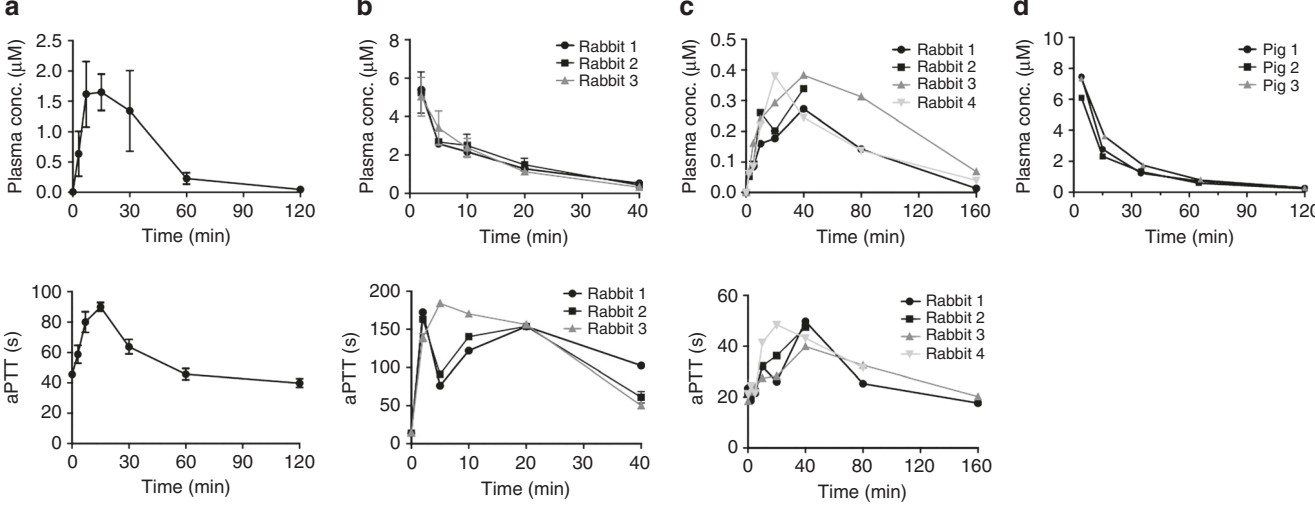

**Fig. 3 Pharmacokinetics of FXII900 in mouse, rabbit, and pig. a** Pharmacokinetics in mouse after subcutaneous administration (5 mg kg$^{-1}$, $n = 3$). Concentration of FXII900 in plasma (upper panel) and aPTT (lower panel) are indicated. Means ± SD are indicated. **b** Pharmacokinetics in rabbit after intravenous administration (3.7 mg kg$^{-1}$, $n = 3$). Concentration of FXII900 in plasma (upper panel) and aPTT (lower panel) are indicated. Plasma samples were analyzed in duplicate for rabbit 1 and in triplicate for rabbits 2 and 3. Mean values are indicated. For rabbit 2 and 3, data are presented as mean ± SD. **c** Pharmacokinetics in rabbit after subcutaneous administration (3.7 mg kg$^{-1}$, $n = 4$). Concentration of FXII900 in plasma (upper panel) and aPTT (lower panel) was measured. Plasma samples were analyzed in duplicate. Mean values are indicated for each rabbit. **d** Pharmacokinetics in pig after intravenous administration (4 mg kg$^{-1}$, $n = 3$). Concentrations of FXII900 in plasma were determined in duplicate, and mean values are indicated for each pig. Source data of **a**–**d** are provided as a Source data file.

We further determined the pharmacokinetics of FXII900 in pigs, as this species offers models for indications in which FXII plays a role, such as ischemic reperfusion injury[25,36]. After intravenous administration ($n = 3$, 4 mg kg$^{-1}$), the inhibitor reached a maximal concentration of around 7 μM and was cleared with a $t_{1/2}$ of 36 ± 5 min (Fig. 3d, upper panel). The clearance rate and volume of distribution were 11.8 ± 1.8 ml kg$^{-1}$ min$^{-1}$ and 610 ± 140 ml kg$^{-1}$

(Supplementary Table 6). In all the studies performed with the three species, we did not observe any signs of toxicity or other adverse effects.

**FXII900 inhibits ferric chloride-induced thrombosis in mice.** We next evaluated the thromboprotective properties of FXII900

in a ferric chloride (FeCl$_3$)-induced thrombosis mouse model. When applied directly onto the blood vessels, FeCl$_3$ induces thrombosis by aggregating the red blood cells, which in turn activates platelets at the application site[37–39]. To test the ability of our compound to protect against thrombosis in this model, FXII900 or an inactive control peptide FXII901 were administered subcutaneously to two groups each containing 10 mice (5 mg kg$^{-1}$). The negative control FXII901 is a variant of FXII900 with three modifications (Phe4F3→Phe, Arg4→Ala, βhArg11→Arg) that reduce its inhibitory constant 100,000-fold for FXIIa ($K_i = 39 \pm 14 \mu M$; Supplementary Fig. 8). A solution of 7.5% FeCl$_3$ was applied to the mesenteric arterioles and the blood flow was monitored by intravital microscopy for 25 min. Mice that were injured by FeCl$_3$ could be clearly identified due to a speckled pattern that supposedly represented an accumulation of platelets (Fig. 4a)[40]. Seventeen out of the 20 mice showed this pattern (nine treated, eight control) and were taken for further analysis (Supplementary Figs. 9 and 10). The arterioles of the mice receiving the inactive control peptide FXII901 rapidly formed clots, and in a majority of the animals, the vessels were completely occluded after around 10 min, as exemplified in the lower micrographs in Fig. 4a and shown for all mice in Supplementary Fig. 9. The time point and extent of clot formation observed for the control peptide were as expected based on previous studies in which vessels of non-treated mice were exposed to 7.5% FeCl$_3$[41]. In contrast, mice that were treated with the inhibitor FXII900 showed only the characteristic speckled pattern with the occasional formation of a blood clot (Fig. 4a, upper micrographs; Supplementary Fig. 10). In this group, no complete occlusion was observed. We quantified the anticoagulant effect of FXII900 by comparing the rates for both clot formation (a clearly visible blood clot; Fig. 4b, left panel) and blood vessel occlusion (a blood clot with a diameter of the blood vessel; Fig. 4b, right panel). Of the eight mice receiving the control peptide, seven showed blood clotting and five showed full-vessel occlusion. In contrast, in the FXII900-treated mice, only three showed blood clotting, and in no mice were the vessels fully occluded ($p$ for chi-squared test = 0.02 and 0.005, respectively). In addition, the few mice in the treated group that showed signs of coagulation developed these blood clots on average around 10 min later than the control group ($p$ for chi-squared test = 0.006; Supplementary Table 7).

Repeating the experiment with a 5-fold higher dose of FXII900 ($n = 8$, 25 mg kg$^{-1}$) and the negative control FXII901 ($n = 9$, 25 mg kg$^{-1}$) gave comparable results to the lower dose, with no further delay in coagulation or reduction in the rate of clot formation (Fig. 4b; intravital microscopy for 20 min; by mistake 5 min shorter than in the experiment with lower dose). In this repeat study, a positive control using the current gold-standard drug heparin was added for comparison ($n = 10$, 200 IU kg$^{-1}$). In this control group, more mice showed blood clotting (six) and vessel occlusion (one) than in the two groups treated with FXII900 (Fig. 4b).

**Mice treated with FXII900 have a normal hemostatic capacity.** We assessed the bleeding propensity of mice treated with the higher of the two doses of FXII900 used in the above ferric chloride-induced thrombosis study (25 mg kg$^{-1}$, SC) using a tail transection model (tail transection at 2 mm from the tail end, diameter > 1 mm; $n = 6$). As positive and negative controls, we treated mice with PBS ($n = 6$) and heparin (200 IU kg$^{-1}$, IV, $n = 6$). We placed the tails into a tube with warmed PBS and measured the time of bleeding and the volume of blood lost. Mice injected with heparin bled essentially continuously throughout the monitored 30 min period, which is similar to previous

reports[41] and showed either a small (3 mice) or large loss of blood (3 mice). Given the heterogeneous result for the blood loss, we performed the heparin control with four additional mice, which showed medium to high blood loss. Mice treated with the FXIIa inhibitor showed an average bleeding time of 10 min and blood loss of 21 μl, which was comparable to the PBS-treated mice (average bleeding time: 7 min, average blood loss: 34 μl) and the FXII901 control (Fig. 4c, Supplementary Fig. 11), which indicated that the FXIIa inhibitor does not affect the hemostatic capacity.

**FXII900 provides bleeding-free anticoagulation in artificial lungs.** To test the clinical potential of FXII900 for inhibiting FXII-driven coagulation in ECMO, we tested the inhibitor in an artificial lung rabbit model. New Zealand White rabbits were anesthetized, either treated intravenously with FXII900 ($n = 4$, 2 mg kg$^{-1}$ bolus and 0.075 mg kg$^{-1}$ min$^{-1}$ infusion for four hours), left untreated ($n = 3$), or treated with heparin for comparison ($n = 2$, infusion of 60 IU h$^{-1}$ and rate adjustments to maintain ACT within a clinical range of 220–300 s), and were mounted for a 4-h period to a veno-venous ECMO configuration using a polymethylpentene (PMP) fiber artificial lung system.

Treatment with FXII900 prolonged the coagulation parameters aPTT and activated clotting time (ACT) around 10-fold during the entire course of the experiment. This was to a much larger extent than those for heparin, which were prolonged 1.5 and 2-fold, the range recommended for heparin anticoagulation (Fig. 5a and b). For most aPTT measurements using FXII900, no coagulation was observed at 240 s, the maximal aPTT value measured with this device (Supplementary Fig. 12a). In untreated rabbits, the coagulation times remained at baseline values of around 20 s (aPTT) and 170 s (ACT) throughout the experiment. Because increased pressure at the inlet of the artificial lung indicates clogging that can be caused by blood coagulation, we measured the resistance of the artificial lung, indicated by the difference in pressure at the inlet and outlet of the lung divided by the flow. In all rabbits treated with FXII900, the resistance remained at the baseline of 50 mmHg L$^{-1}$ min$^{-1}$ during the entire experiment, except for one rabbit in which the resistance started to increase strongly after two hours at baseline (Fig. 5c and Supplementary Fig. 12c). This was in stark contrast to the untreated animals in which the resistance increased in the first hour and doubled for all rabbits from 50 mmHg L$^{-1}$ min$^{-1}$ to 100 mmHg L$^{-1}$ min$^{-1}$ during the course of the experiment.

Because excessive bleeding is a major concern with standard anticoagulation in ECMO systems, we analyzed if FXII900 affected the hematological parameters of bleeding time and platelet count. Rabbits treated with the inhibitor showed completely normal bleeding times throughout the experiment (4–5 min; Fig. 5d and Supplementary Fig. 12d). In contrast, the group treated with heparin at a clinically relevant dose showed 2-fold prolonged bleeding times (5–10 min) at all the time points measured (Fig. 5d and Supplementary Fig. 12d). The platelet count was essentially the same in the three groups, decreasing around two-fold from an initial level of $2 \times 10^8$ cells per ml (before connecting the lung) and remaining constant over the entire course of the experiment (Fig. 5e and Supplementary Fig. 12e). Analysis of the artificial lungs at the end of the experiment showed that the inhibitor- and heparin-treated rabbits had a reduced amount of clotted blood. The volume of the clot in relation to the device volume was $10 \pm 6\%$ for the FXII900-treated rabbits, which was significantly smaller than that of untreated rabbits ($37 \pm 10\%$; $p = 0.03$), but it was slightly larger than that of the heparin-treated rabbits (5%; Fig. 5f and Supplementary Fig. 12f).

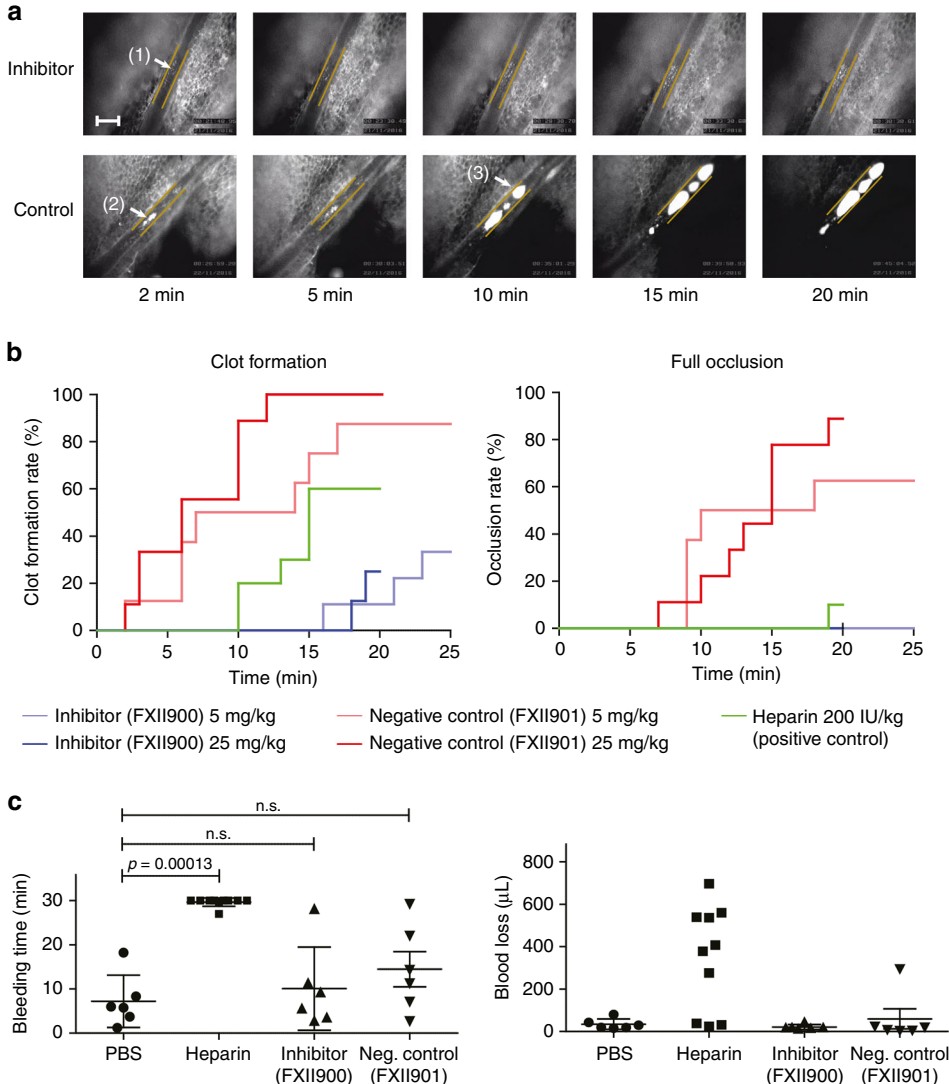

**Fig. 4 Thromboprotection of FXII900 in mice and effect on bleeding. a** Intravital fluorescence microscopy images showing mesenteric arterioles in which thrombosis was induced by topical application of $FeCl_3$ (7.5%, 1 min). Platelets were fluorescently labeled with Rhodamine 6G for visualization. Representative images are shown for two mice, one treated with FXII900 (upper panels) and one treated with inactive control peptide FXII901 (lower panels) 15 min before the application of ferric chloride (5 mg kg$^{-1}$, subcutaneous injection). Vessel walls at the $FeCl_3$ application site are indicated with yellow markers. Three distinct morphological changes, (1) a characteristic speckled pattern, (2) clot formation, (3) and vessel occlusion, are indicated. Scale bar: 200 μm. **b** The percentage of mice showing either clot formation or full occlusion at different time points after ferric chloride application is indicated over time (inhibitor FXII900: 5 mg kg$^{-1}$, $n = 9$; 25 mg kg$^{-1}$, $n = 8$; neg. control FXII901: 5 mg kg$^{-1}$, $n = 8$, 25 mg kg$^{-1}$, $n = 9$; heparin: $n = 10$). Clot formation was defined as the appearance of an aggregate with a diameter larger than 10 μm. Full occlusion of the blood vessel was defined as a blood clot having the same diameter as the blood vessel. **c** Blood loss and bleeding time of mice with 2 mm tail transections. Mice were treated 5 min before clipping the tail tips with the vehicle (PBS, IV, $n = 6$), heparin (200 IU kg$^{-1}$, IV, $n = 10$), FXII900 (25 mg kg$^{-1}$, SC, $n = 6$) or FXII901 (25 mg kg$^{-1}$, SC, $n = 6$). Mean values and standard deviations are indicated. The significance for a prolonged bleeding time was assessed with an unpaired, one-tailed *t*-test. The *p* value is indicated for significant results and n.s. stands for not significant. Mice treated with heparin showed either a small or medium-to-large loss of blood, and a mean value is thus not indicated.

## Discussion

Various protein-based inhibitors were developed that showed that suppression of FXIIa activity provides efficient anti-coagulation without increasing bleeding risks, though the development of high-affinity small-molecule FXIIa inhibitors has been lagging. Given the attraction of small, synthetic molecules as therapeutics, in this work, we improved the binding affinity and stability of FXII618, a peptide macrocycle FXIIa inhibitor that was previously developed by phage display. The generated inhibitor FXII900 blocks FXIIa with a $K_i$ of 370 pM, shows 100,000-fold selectivity over other plasma proteases, and is stable in blood plasma.

The improvement of the binding affinity of FXII618 by 21-fold and proteolytic stability by 25-fold was achieved by combining random screening and rational design strategies. Two amino acid substitutions that enhance the affinity and/or stability of the inhibitor were identified by screening a panel of singly mutated peptides, and were combined with two previously identified beneficial mutations. Individual reversion of the amino acid substitutions in FXII900 showed that each single one contributes to improving the binding affinity and that the modifications of Arg1 to D-Arg-D-Ser and Arg11 to βhArg are the main ones that improve the stability. The higher binding affinity translates into a prolongation of the aPTT in all species tested, with the $EC_{1.5x}$

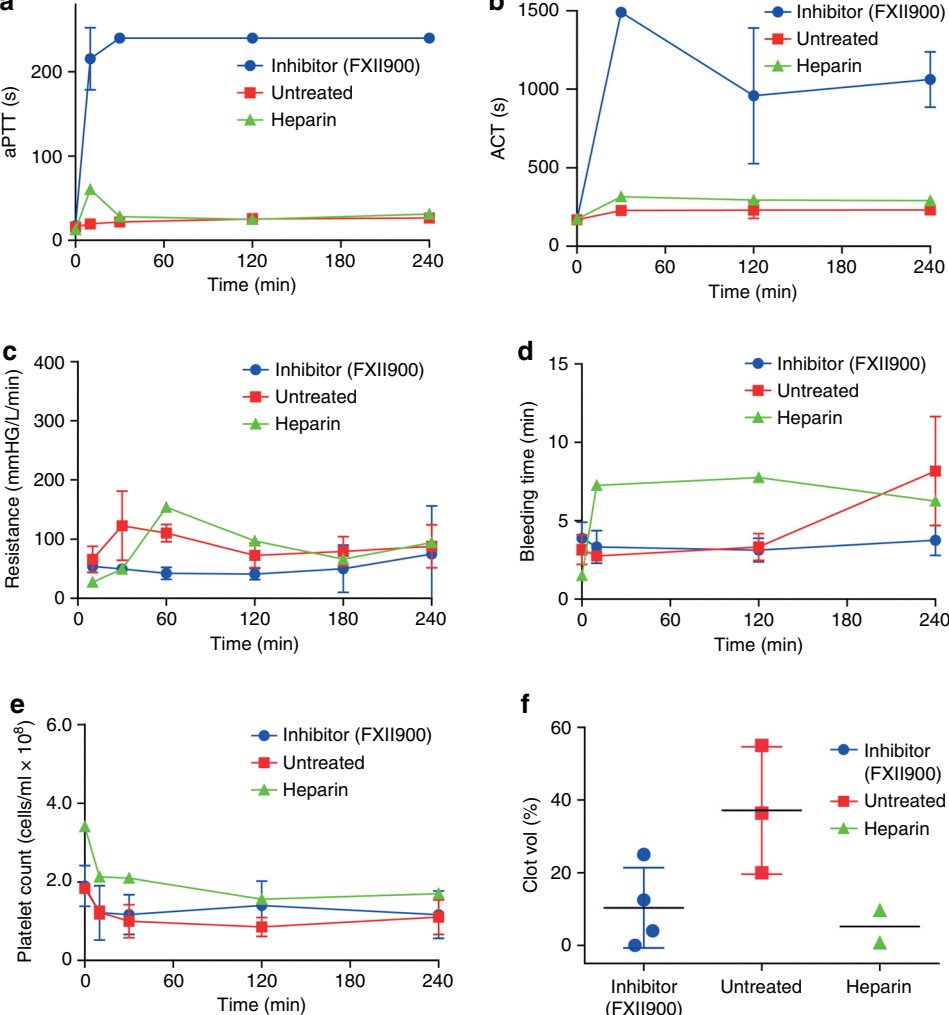

**Fig. 5 Bleeding-free anticoagulation in an artificial lung model in rabbits.** Rabbits were connected veno-venous to an artificial lung system for four hours. FXII900 was injected as a bolus (2 mg kg$^{-1}$) before the start of the extracorporeal circulation and as a constant infusion (0.075 mg kg$^{-1}$ min$^{-1}$) over the full time course of the experiment ($n = 4$; data shown in blue). Rabbits in the control group were untreated ($n = 3$, data shown in red). Rabbits in the heparin group ($n = 2$) were initially infused with one IU per min heparin and the drip rate was adjusted to maintain ACT values between 220 and 300 s. Values indicated for the time point 0 refer to measurements made before connecting the artificial lung. Mean ± SD is indicated for the four rabbits in the FXII900-treated group and the three rabbits in the control group. Mean values are indicated for the two rabbits in the heparin group. Data from individual rabbits is shown in Supplementary Fig. 12. **a** aPTT. Coagulation times of 240 s indicate that plasma did not coagulate at this time. **b** ACT. Clotting times of 1500 s indicate that blood did not coagulate at this time. **c** Resistance calculated based on pressure at the inlet and outlet of the device and the flow rate. **d** Bleeding time measured by incision-provoked injuries at the ears. **e** Platelet count. **f** Volume of blood clots indicated in % of volume of the artificial lungs.

improved 4.3-fold in human plasma and 10-fold in mouse plasma. As expected, the EC$_{1.5x}$ values are much higher than the $K_i$ values because FXII needs to be inhibited nearly fully to prolong aPTT 1.5-fold (we estimate > 99%) while the $K_i$ is by definition the concentration at which 50% of FXII is inhibited. The much-enhanced potency in mouse plasma was of interest in view of the planned pharmacologic study in mice. FXII900 did not prolong PT at 120 μM at all, which contrasted with FXII618 that prolonged the PT slightly at this highest concentration tested, indicating that modification of the four amino acids had also improved the selectivity of the inhibitor. The enhancement of affinity, selectivity, and stability achieved in this work is owed mostly to unnatural amino acids that could be incorporated due to the synthetic nature of the inhibitor, which is an advantage over protein-based inhibitors that need to be produced by recombinant expression and do not easily allow the incorporation of unnatural building blocks.

FXII900 is composed of nine natural amino acids, four non-natural ones, and a central core structure that connects the side chains of three cysteines. The inhibitor can be conveniently synthesized by automated solid-phase peptide synthesis in large quantities. All applied non-natural amino acids are inexpensive, allowing for production of inhibitor for a moderate cost. Crude linear peptide was cyclized efficiently by the chemical linker TATA and only a single HPLC purification step was required to obtain > 95% pure product with isolated yields greater than 50%. We provide a supplementary figure that describes the exact peptide sequence and the conditions recommended for peptide cyclization, which may be used for ordering the peptide from companies offering custom peptide synthesis (Supplementary Fig. 3).

FXII900 is rapidly cleared upon intravenous administration, most likely by renal filtration, as expected based on the small size and the polar structure. The elimination half-life ranges from

12 min in rabbits to 36 min in pigs. The prolongation of the aPTT correlated with the PK data for all species. Based on allometric scaling, the half-life in humans can be expected to be in the range of one hour. No degradation products were detected in plasma samples of all species, suggesting that the inhibitor fully resists proteases in the blood circulation and that it is filtered out as an intact molecule, which is in line with the high stability and long half-life in plasma found ex vivo. Despite the fast clearance, a single-dose injection of FXII900 prolonged the aPTT in mice and rabbits for 30 min. The relatively short circulation time of these small molecules, especially as opposed to protein therapeutics, allows flexible adjustment of the duration for which the body is exposed to the inhibitor in therapeutic applications. The pharmacokinetic profile of FXII900 makes it attractive for acute disease treatment such as CPB, hemodialysis and ECMO, or anticoagulation for short time periods after ischemic stroke or surgery, wherein a patient is hospitalized and could be given infusions of the inhibitor for the desired time period. For hemodialysis applications, the inhibitor would potentially need to be tethered to a large protein or polymer to prevent rapid removal by the filtration unit. For controlling the activity of FXII over several days, weeks or months, FXII900 is not suited and other therapeutic modalities such as antibody-based FXII inhibitors or antisense oligonucleotides are clearly better options. Given its high proteolytic stability, FXII900 could potentially resist to some extent proteases of the stomach and intestines, but due to the multiple charges, it would unlikely be able to cross the gastro-intestinal epithelium, preventing oral administration.

We evaluated the pharmacologic activity of FXII900 in a ferric chloride-induced thrombosis mouse model. FXII-deficient mice show strong reduction of coagulation in this model, indicating that FXII plays a central role in this process[8,9]. We found that a bolus injection of FXII900 efficiently protected mice from thrombosis, while bleeding time and blood loss were not affected, as assessed in a tail transection bleeding model. This was in strong contrast to a control experiment with heparin (200 IU kg$^{-1}$) in which mice were constantly bleeding throughout the 30 min period monitored and lost a large volume of blood. It is important to say, however, that thrombosis in myocardial infarction, ischemic stroke, or pulmonary embolism is more complex than in the applied mouse thrombosis model, as thrombosis can also be caused by triggers that are not dependent on FXII activation. Independent of this, the results obtained show that a synthetic FXIIa inhibitor can efficiently suppress coagulation in cases where it is mediated via FXII activation.

Finally, we further tested if FXII900 could suppress the coagulation activated by the membrane of artificial lungs using a rabbit model. Heart-lung machines are used for heart surgery because of the difficulty of operating on the beating heart such that approximately half a million cardiac operations performed per year in the US use CPB. Similar, more compact systems are also used to support patients with cardiac and respiratory dysfunction, with more than 7000 patients treated annually with extracorporeal life support[42]. The contact of blood with the hydrophobic polymer surfaces found in these machines induces FXII contact activation and thus coagulation. This coagulation is currently suppressed using high doses of heparin, which bears bleeding risks. An analysis of nearly 80,000 ECMO patients showed that hemorrhage is the major adverse event in these situations, with bleeding from surgical sites occurring in 6.3% to 29.3% of patients depending on the patient group[42]. Moreover, both device thrombosis and patient bleeding lead to significant increases in patient mortality[43]. We found that continuous infusion of FXII900 efficiently prolonged the coagulation parameters ACT and aPTT over four hours and thus the entire course of the experiment. The inhibitor reduced coagulation in an artificial lung as measured by the resistance of the device, which remained low and indicated reduced clogging by coagulation and a reduction in the total volume of the blood clots in the device, which was more than three-fold lower in treated rabbits. Importantly, all rabbits treated with FXII900 showed normal bleeding times. These findings indicated that a small molecule FXIIa inhibitor is suitable for reducing blood coagulation in CPB. While heparin may still be required to suppress coagulation triggered through routes other than FXII, the application of a small molecule FXIIa inhibitor may allow for a reduced heparin dose, which would reduce the bleeding risks. In addition to its role in thrombosis in ECMO and CPB, FXIIa initiates the inflammatory kallikrein-kinnin system[19]. Bradykinin plasma concentrations are largely elevated in CPB and ECMO patients, so targeting FXIIa may provide an additional anti-inflammatory benefit to these patients, to reduce or prevent adverse effects, such as organ damage.

In conclusion, a synthetic FXIIa inhibitor with sub-nanomolar affinity, high selectivity, and good stability has been developed that allows for efficient anticoagulation in relevant animal disease models without increasing bleeding risks, addressing several concerns surrounding the current gold-standard treatment of heparin. The synthetic nature and the small size allows for efficient production of the compound. With its excellent binding properties and stability as demonstrated herein, the inhibitor may be readily applicable in its current form for acute procedures or conditions associated with an increased risk of thrombosis, such as CPB during heart surgery or ECMO without compromising hemostasis at the wound site.

## Methods

**Study design**. The objectives of this study were to improve the pharmacologically relevant properties of a peptide macrocycle inhibitor of FXIIa, to assess the pharmacokinetic properties of the resulting inhibitor in three species, and to evaluate the inhibitor in two clinically relevant animal models. The inhibitor was engineered by synthesizing variants of the lead peptide and testing their inhibitory activity, specificity, and stability in various assays in vitro. Inhibitors showing improved activities were prepared with greater than 95% purity and their activities were tested at least in duplicate. The affinity of the final peptide FXII900 was determined in quintuplicate from three different synthetic batches. The pharmacokinetic properties in mice, rabbits, and pigs were assessed by quantifying the inhibitor in plasma samples using liquid chromatography and mass spectrometry, and by measuring aPTT in plasma samples in duplicate or triplicate, as indicated. The pharmacologic effects of the inhibitor were tested in a FeCl$_3$-induced thrombosis model in mice and in an artificial lung model in rabbits. In all animal experiments, subjects were randomly assigned to groups.

**Screening method based on coumarin-labeled peptides**. In order to test the inhibitory activity of peptides without prior purification, we developed the following method. Variants of the lead FXII618 peptide were synthesized with a C-terminal amino acid Fmoc-β-(7-methoxy-coumarin-4-yl)-Ala-OH. The fluorophore 7-methoxy-coumarin allowed precise quantification of the crude peptide. After solid-phase peptide synthesis, cleavage, and ether precipitation, the peptides were chemically cyclized and their $K_i$ values were determined in a FXIIa activity assay using a fluorogenic substrate. Peptides synthesized and characterized with this strategy displayed an around 2.5-fold higher apparent $K_i$ than analogous HPLC-purified peptides without the tag. For example, the reference peptide FXII618 carrying the coumarin amino acid had an around 2.5-fold weaker $K_i$ (20 ± 3 nM) compared to the purified FXII618 without the tag (8.1 ± 0.7 nM). Despite the difference, this method allowed for a comparison between peptides and the identification of the most active variants of FXII618. The best peptides from the screen were synthesized without the coumarin label, purified, and characterized.

**Plasma stability assays**. Peptide (2 μl of 2 mM in H$_2$O) was added to 398 μl of citrated human plasma (Innovative Research) to obtain a final peptide concentration of 10 μM. The mixture was incubated in a water bath at 37 °C. At different time points (0, 0.5, 1, 2, 4, 8, 12, 24, 48, and 96 h), samples of 30 μl were removed, diluted to 200 μl with aqueous buffer (10 mM Tris-HCl, pH 7.4, 150 mM NaCl, 10 mM MgCl$_2$, 1 mM CaCl$_2$), and incubated for 20 min at 65 °C to inactivate plasma proteases. The peptide/plasma samples were stored at −20 °C until the residual inhibitory activity of the peptides was measured in a FXIIa inhibition assay. For the activity assay, the peptide/plasma samples were centrifuged for 5 min at 16,000 × g, serial two-fold dilutions of the supernatant were prepared (peptide

concentration ranges from 0.5 nM to 0.5 μM), and the residual activity of 0.5 nM human β-FXIIa was measured using 50 μM Boc-Gln-Gly-Arg-AMC substrate. $IC_{50}$ values were derived from the fitted curve using an equation described in the Supplementary Methods. Residual inhibition in % was calculated using the equation $IC_{50,0h}/IC_{50,xh}*100$, wherein $IC_{50,0h}$ is the functional strength of the inhibitor at time point 0 and $IC_{50,xh}$ the functional strength of the inhibitor after one of the different plasma incubation periods mentioned above.

**Plasma degradation assays.** Plasma stability and peptide cleavage sites were assessed by incubating the bicyclic peptides in mouse plasma and analyzing the products with an LC-MS system (LCMS-2020, Shimadzu). Peptide (2 μl of 2 mM in $H_2O$) was added to 48 μl of citrated mouse plasma (Innovative Research) to obtain a final peptide concentration of 80 μM. The mixture was incubated in a water bath at 37 °C. At different time points (0, 15, 24, 72, 96, 120, and 144 h), samples of 5 μl were removed, mixed with 5 μl of 6 M guanidinium hydrochloride, and incubated for 30 min at RT. Plasma proteins were precipitated by incubation with 200 μl of ice cold EtOH and 0.1% (v/v) formic acid for 30 min and centrifuged at $9000 \times g$ for 20 min at 4 °C. The supernatant was evaporated in a Speedvac at 50 °C and reduced pressure. The residue was dissolved in 40 μl of deionized water containing 0.1% (v/v) CHOOH and analyzed by LC-MS. The samples were analyzed using an analytical C18 column (Phenomenex C18 Kinetex column, 50 × 2.1 mm, 2.6 μm, 100 Å) and a linear gradient of 5–35% solvent B (MeCN, 0.05% [v/v] CHOOH) in solvent A ($H_2O$, 0.05% [v/v] CHOOH) in 5.5 min at a flow of 1 ml per min. The masses of the intact peptide and degradation products were measured on a single quadrupole mass spectrometer in positive ion mode using electrospray ionization. Peptides were quantified based on the absolute intensities of the detected mass peaks ($M^{3+}$ and $M^{4+}$).

**Cloning of vectors for expression of FXII in mammalian cells.** The protein sequences for human, rabbit and pig factor XII were taken from the following database entries: human FXII: UniProtKB - P00748; rabbit FXII: NCBI RefSeq - XP_008253687.1; pig FXII: UniProtKB - O97507. The sequences are shown in the Supplementary Methods. DNA encoding the full-length proteins were ordered from Eurofins Genomics. The codons in these sequences were optimized for mammalian expression using the codon optimization tool from Integrated DNA Technologies (IDT). In addition to the FXII gene, the ordered DNA sequences encode a C-terminal GSGS-linker, a His6-tag and a stop codon, and they are flanked by NheI (GCTAGC) and a HindIII (AAGCTT) restriction sites. The entire DNA sequences are provided in the supporting materials. The DNA sequences were cloned into the pEXPR-IBA42 vector downstream of a BM40 signal sequence for secreted expression in mammalian cells. The ligated vector was transformed into DH5 alpha electrocompetent *E. coli* cells. Plasmid DNA from single clones was sequenced by Sanger sequencing.

**Recombinant expression, purification, and activation of FXII.** 1.5 mg plasmid DNA was transfected into 500 ml CHO cells (Thermo Fisher Scientific) in cell suspension culture using polyethylenimine (PEI). Cells were incubated for seven days at 37 °C, 5% $CO_2$ under shaking conditions. The cells were removed by centrifugation and the secreted protein was purified from the supernatant using a nickel-charged immobilized metal affinity chromatography (IMAC) column (5 ml HisTrap FF Crude, GE Healthcare). The column was equilibrated with buffer containing 15 mM imidazole, 100 mM NaCl, 20 mM Tris-HCl, pH 7.4. The pH of the cell culture supernatant was adjusted to 8.0 using NaOH. The supernatant was run through the column at a flow of 5 ml per min. The column was washed with 20 column volumes of equilibration buffer at a flow of 5 ml per min. The protein was eluted with 500 mM imidazole, 100 mM NaCl, 20 mM Tris-HCl, pH 7.4 at a flow of 5 ml per min. The buffer was subsequently exchanged to 100 mM NaCl, 20 mM Tris-HCl, pH 7.4 by three iterative steps of 10-fold dilution and 10-fold concentration using a 10,000 MWCO centrifugal concentrator. The concentration was determined by measuring absorption at 280 nm (ε = 100,000). For the activation of recombinant FXII, 10 μg protein was diluted to a total volume of 10 μl 100 mM NaCl, 20 mM Tris-HCl, pH 7.4 and further diluted with 10 μl of 2-fold concentrated assay buffer (300 mM NaCl, 20 mM MgCl₂, 2 mM CaCl₂, 20 mM Tris-HCl, 0.02% (v/v) Triton-X100, pH 7.4). Dextran sulfate 500 kDa (DXS500) was added to a final concentration of 0.2 μg μl⁻¹ and incubated for 1 hour at 37 °C. 5 μg protein was analyzed by SDS-PAGE under reducing conditions.

**Animal study authorization.** All experiments in mice and pigs were conducted in accordance with the terms of the Swiss animal protection law and were approved by the animal experimentation committee of the cantonal veterinary service (Canton of Berne, Switzerland). The pharmacokinetic studies in rabbits were performed at Washington Biotech Inc. following ethical standards for animal studies of the Office for Laboratory Animal Welfare (OLAW), a division of the US Public Health Service as administered by the National Institutes for Health. The extracorporeal circulation studies in rabbits were performed in compliance with the Allegheny Health Network Institutional Animal Care and Use Committee. The studies in rabbits were approved by the American Association for Laboratory Animal Science (IACUC).

**Pharmacokinetic study in mice.** Mice for all experiments were kept at ambient temperature (around 20 °C), around 60% humidity, half-day light/half-day dark cycles, and in groups of 2–5 mice per cage. C57BL/6J wild-type mice (male, 10–20 weeks old, 25–30 g, Charles River) were injected subcutaneously over the shoulders with 5 mg kg⁻¹ of FXII900 (0.5 mg ml⁻¹ in PBS, pH 7.4). The mice were anaesthetized 3 min before the scheduled blood collection time point (40 mg kg⁻¹ pentobarbital). An abdominal midline incision was performed and 450 μl of whole blood was drawn from the inferior vena cava into a syringe containing sodium citrate (50 μl 3.2% [w/v] sodium citrate). The mice were euthanized by cervical dislocation. The blood was immediately processed to plasma by centrifugation for 10 min at $2000 \times g$ and 4 °C, and stored at −80 °C. The concentration of FXII900 in the plasma samples was determined by LC-MS as described below. Different mice were used for each time point.

**Pharmacokinetic study in rabbits.** The pharmacokinetic properties of FXII900 applied intravenously were determined as follows[44]. Female New Zealand White rabbits (10–20 weeks old) were injected with 3.7 mg kg⁻¹ FXII900 dissolved in 1 ml PBS, pH 7.4 via the ear vein. Blood samples (2.7 ml) were collected at different time points into sodium citrate tubes (BD Vacutainer ref # 363083) and immediately processed to plasma by centrifugation at $1400 \times g$ for 15 min. The concentration of FXII900 in the plasma samples was determined by LC-MS as described below. The pharmacokinetic properties of FXII900 applied subcutaneously were determined as follows. Female New Zealand White rabbits (2.5–2.9 kg) were injected subcutaneously over the shoulders with 3.7 mg kg⁻¹ of FXII900 dissolved in 1 ml PBS pH 7.4. Blood samples (1.8 ml) were collected at different time points into sodium citrate tubes (BD Vacutainer ref # 363080) and immediately processed to plasma by centrifugation at $1800 \times g$ for 10 min. The plasma was stored at −80 °C. The concentration of FXII900 in the plasma samples was determined by LC-MS as described below.

**Pharmacokinetic study in pigs.** Swiss large white pigs, both sexes, 3–4 months old (30 ± 5 kg) were anesthetized and prepared following procedures used to study myocardial ischemia/reperfusion injury[45]. In this procedure, the ACT is monitored and 2500 IU of heparin are injected when ACT values fall below 180 s. Pigs were injected intravenously with 4 mg kg⁻¹ of FXII900 dissolved in 1 ml PBS, pH 7.4. Blood samples (2.9 ml) were collected at different time points into sodium citrate tubes (Sarstedt S-Monovette ref # 04.1902.001) and immediately processed to plasma by centrifugation at $1400 \times g$ for 15 min at RT. The concentration of FXII900 in the plasma samples was determined by LC-MS as described below.

**Quantification of inhibitor in plasma samples.** The concentration of FXII900 in the plasma samples was quantified based on peak intensities of total ion current (TIC) chromatograms acquired by LC-MS (LCMS-2020, Shimadzu). To 15 μl of plasma sample, 1 μl of internal standard peptide and 5 μl of 6 M guanidinium hydrochloride solution were added and mixed. Plasma proteins were precipitated by the addition of 400 μl of ice cold ethanol (99.9% [v/v] EtOH, 0.1% [v/v] TFA) and incubated on ice for one hour. Precipitate was removed by centrifugation ($9000 \times g$, 20 min, 4 °C) and the supernatant dried by centrifugal evaporation under vacuum. Dried samples were dissolved by sequentially adding 2 μl of DMSO and 18 μl of $H_2O$ containing 0.1% (v/v) CHOOH and analyzed by LC-MS. The samples were analyzed using an analytical C18 column (Phenomenex C18 Kinetex column, 50 × 2.1 mm, 2.6 μm, 100 Å) and a linear gradient of 5–30% solvent B (MeCN, 0.05% [v/v] CHOOH) in solvent A ($H_2O$, 0.05% [v/v] CHOOH) in 4.5 min at a flow of 1 ml per min. The mass was measured on a single quadrupole mass spectrometer in positive ion mode using electrospray ionization. Peptides were quantified based on the absolute intensities of the detected mass peaks ($M^{3+}$ and $M^{4+}$).

**FeCl₃ injury thrombosis model in mesenteric arterioles in mice.** A model of thrombosis in mesenteric arterioles using intravital microscopy was performed according to Angelillo-Scherrer A. et al.[46] with minor modifications. C57BL/6J wild-type mice (male, 10–20 weeks old, 25–30 g, Charles River) were injected intravenously with Rhodamine 6G (100 μl, 1 mM, ACROS Organics product 41902) to fluorescently label the platelets and leucocytes. The mice were injected subcutaneously over the shoulders with FXII900 (5 or 25 mg kg⁻¹, 0.5 mg ml⁻¹ in PBS, pH = 7.4) or the negative control FXII901 (5 or 25 mg kg⁻¹, 0.5 mg ml⁻¹ in PBS, pH = 7.4), or intravenously (retro-orbital) with 200 IU kg⁻¹ heparin, and subsequently anesthetized with ketamine (80 mg kg⁻¹) and xylazine (16 mg kg⁻¹) via intraperitoneal injection. An abdominal midline incision was made to expose the mesenteric arterioles which were imaged by intravital microscopy using a Mikron IVM500 microscope (Mikron Instruments) coupled with a 50 W mercury lamp (HBO 50 microscope illuminator, Zeiss) attached to combined blue (exciter 455DF70, dichroic 515DRLP, and emitter 515ALP) and green (exciter 525DF45, dichroic 560DRLP, and emitter 565ALP) filter blocks. Thrombus formation was induced by the application of a 1 × 2 mm filter paper saturated with FeCl₃ solution (7.5% [w/v], Roth, art no 5192.1) onto the blood vessel for 1 min. The blood flow, clot formation and vessel occlusion were monitored for 20 or 25 min wherein images were recorded every minute using a digital video cassette recorder (DSR-11, Sony) and analyzed using ImageJ software (version 1.52). Mice were euthanized by

final bleeding and cervical dislocation. Time to clot formation and full occlusion of the blood vessel were determined as follows. Clot formation was defined as the formation of a clear aggregation with a diameter of around 10 µm or larger. Occlusion was defined as a clot covering the full diameter of the vessel. Mice showing no speckled pattern at the site of $FeCl_3$ application were assumed to be not injured at the blood vessel and were excluded from the analysis.

**Tail bleeding time and volume.** Mice (8- to 10-week-old) were anesthetized with pentobarbital (50 mg kg$^{-1}$, IP), treated with PBS (IV, retro-orbital), 200 IU kg$^{-1}$ heparin (IV, retro-orbital), 25 mg kg$^{-1}$ FXII900 (SC) or 25 mg kg$^{-1}$ FXII901 (SC), and put on a 37 °C heating pad. After 5 min, the distal tail was transected at 2 mm with a disposable surgical blade, and the diameter measured to confirm that it is > 1 mm. The tail was placed in a 50 ml falcon tube filled with phosphate-buffered saline (warmed to 37 °C) and the bleeding time was recorded. The total blood loss was determined by measuring the absorption at 540 nm.

**Artificial lungs.** Mini-lungs were created that allowed for blood exposure to the fibers in the artificial lung, but for construction simplicity, do not transfer gas. Polymethylpentene (PMP) fiber (3 M) with 50% porosity, a 380 µm outer diameter, and two layers at 30° cross angles were cut into 1.78 × 1.78 cm squares. Five of these layers were put together, making sure the fibers all ran in the same direction, and they were sealed together into a fiber bundle using a hot plate on each side of the square sheets. These fiber bundles were melted such that they had a 1.57 × 1.57 cm square frontal area perpendicular to the flow. Eight 5-layer "chiclets" were placed together to make up the fiber bundle of one mini lung device, giving a final surface area of 263 cm$^2$. These fiber bundles were placed in a square plastic housing of 3.05 cm in length, making sure that the fiber bundles fit tightly in the device to prevent shunting around the fiber bundle. This housing was attached to plastic end caps that were placed on each end with a 1/8″-barbed tube fitting for 3/16″-ID tubing to allow the device to be connected to tubing in the circuit. These end connectors were coated with Teflon tape before connecting them to the plastic end caps to prevent leakage. The device was held together with two screws going from end cap to end cap. The entire device was secured with silicone to eliminate leakage. The silicone was left to dry for 24 h, and the device leak tested as follows. Filtered deionized water was run through the device to check for leaks within the device. If no leaks were found, the device was left to dry with filtered air running through the device for 24 h.

**Extracorporeal circulation in rabbits.** New Zealand white rabbits of 3.2–4.2 kg (Charles River) were anesthetized via intramuscular injections of ketamine (30 mg kg$^{-1}$) and xylazine (5 mg kg$^{-1}$). One of the ear veins was catheterized via a 24G winged catheter, and the rabbits were intubated with a 3.0 endotracheal tube. The animals were kept anesthetized throughout the four-hour experiment via inhaled isoflurane (2%) and were ventilated with a peak inspiratory pressure (PIP) of < 20 cmH$_2$O, positive end expiratory pressure of 5 mmHg, tidal volume of 4–6 ml kg$^{-1}$, and a respiratory rate of 22–60 breaths per min. Tidal volume and respiratory rate were adjusted to maintain normal arterial blood gases and the listed PIP. Phenylephrine was applied intravenously at a rate of 0.5–5 µg kg$^{-1}$ min$^{-1}$ to maintain blood pressure. For monitoring blood pressure and collecting blood samples, the rabbits' right or left carotid arteries were cannulated using a 16-gauge angiocatheter (Becton Dickinson) and were secured with silk ties. The device and circuit were first primed with filtered CO$_2$ and then with saline (NaCl, 0.9% [w/v]) containing solumedrol (30 mg kg$^{-1}$). At this point, rabbits of the inhibitor-treated group were injected with 2 mg kg$^{-1}$ of FXII900 as a 2 mM solution in PBS (pH 7.4) via the ear vein and the circuit was connected to the rabbits via a venous/venous ECMO configuration using a 14-gauge angiocatheter (Becton Dickinson) and a 6″ pressure tubing that was cut to length from a 24″ pressure tubing (Edwards Lifesciences) in the right and left internal jugular veins, respectively. The circuit was placed in a roller pump (Cobe), and the blood flow was set to 45 ml per min. After connecting the extracorporeal circuit, rabbits of the inhibitor-treated group were infused with 0.075 mg kg$^{-1}$ min$^{-1}$ FXII900 via the ear vein for the entire duration of the study. Rabbits treated with heparin were initially infused with 60 IU h$^{-1}$ (100 IU ml$^{-1}$) starting before connection to the circuit. The drip rate was adjusted based on the following nomogram to maintain the ACT in a range between 220 and 300 s. ACT > 320 s: decrease infusion rate by 12 IU h$^{-1}$, 320 s > ACT > 300 s: decrease infusion rate by 6 IU h$^{-1}$, 300 s > ACT > 220 s: maintain heparin infusion rate, 220 s > ACT > 200 s: increase infusion rate by 6 IU h$^{-1}$, 200 s > ACT > 180 s: increase infusion rate by 12 IU h$^{-1}$, ACT < 180 s bolus 100 U (1 ml). After four hours, the animals were euthanized by potassium chloride (2 mg kg$^{-1}$, IV). At the end of the experiment, 5 ml of heparin was run through the artificial lung while it was still connected to the rabbits. Then the circuit was removed, and the device was washed with saline carefully so that no clot shedding occurred. Saline was run through the device until the drained portion the effluent was clear. Clot volume was measured by measuring the volume of the device prior to the experiment and again at the end of the experiment after washing. This was done by completely filling the device with saline and recording this amount of saline as the device volume. The difference between the beginning and ending volume was determined to be the clot volume. From the clot volume, the percent of clot within the device was calculated.

**Data acquisition during extracorporeal circulation.** Platelet and white blood cell counts, hematocrit, arterial blood gases (ABG), ACT, aPTT, fibrinopeptide A (FPA), device resistance, and bleeding time were measured prior to circuit attachment and at 10, 30, 120, and 240 min following the initiation of ECMO. For the platelet counts, a syringe with 0.05 ml of 3.2% sodium citrate (w/v) was used to draw 0.45 ml of blood for a total volume of 0.5 ml. This was then centrifuged at 60 × g for 10 min, and 20 µl of the plasma was placed in 20 ml of ISOTON® diluent and counted using a Coulter Counter (Beckman Coulter, Inc. Brea, CA) with a 50 µl aperture tube. For counting, cells were considered platelets if their diameters were 1.8–5.6 µm. For white blood cell counts, 40 µl of whole blood was placed in 20 ml of the ISOTON® diluent. Six drops of ZAP-OGLOBIN lysing solution were added to the mixture and mixed gently. This was allowed to sit for 2 min. The white blood cells were also counted using the Coulter Counter (Beckman Coulter) with a 50 µl aperture tube where any particle above 3.6 µm in diameter was considered a white blood cell. Arterial blood gases were measured by drawing 0.4 ml of blood into a heparinized syringe and run using an arterial blood gas analyzer (ABL800 FLEX, Radiometer). The ACT and hematocrit were measured by collecting 0.5 ml of blood. The ACT was measured using a Hemochron analyzer with tubes containing glass beads as the activator. The hematocrit was measured via capillary centrifugation. For aPTT measurements, blood samples (1.8 ml) were collected at different time points into tubes containing sodium citrate (0.2 ml, 3.2% [w/v] sodium citrate) and immediately processed to plasma by centrifugation at 2000 × g for 15 min at 4 °C. The samples were then analyzed as described above. All platelet counts, WBC counts, FPA, and FXIIa levels were corrected for hemodilution by adjusting the raw values based on the hematocrit. The inlet and outlet pressure and blood flow rate were measured using a Biopac system (Aero Camino Goleta, CA) and pressure transducers at the inlet and outlet of the device (Edwards Lifesciences), and the resistance was calculated with the standard $R = (P_i − P_o)/Q$ where $P_i$ is the inlet pressure in mmHg, $P_o$ is the outlet pressure in mmHg, and $Q$ is the flow in L min$^{-1}$. The bleeding time was measured by cutting small incisions of 4–5 mm at different sites of the right or left ear in each animal. Blood from the incision was removed with gauze every 30 s, and the time until the bleeding stopped was measured.

**Statistical analysis.** For all experiments, mean values are indicated for independent replicates. For experiments performed in triplicate or more, means and standard deviations are indicated. All statistical analyses were performed using GraphPad Prism (version 5) or Microsoft Excel software (version 2016). A chi-squared test was used to determine whether mice were protected against clot formation and full occlusion in the FeCl$_3$ thrombosis model. A one-tailed student's t-test was applied to determine the significance of the difference in the time until clot formation occurred between the two groups. A one-tailed test is appropriate since the effect is only expected in one direction. A one-tailed Student's t-test was applied to assess the significance for the bleeding time prolongation in the mouse tail bleeding experiment.

**Reporting summary.** Further information on experimental design is available in the Nature Research Reporting Summary linked to this paper.

## Data availability
The data supporting the findings of this study are available within the article and Supplementary Information. All datasets generated during the current study are available from the corresponding author on reasonable request. Source data are provided with this paper.

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

## Acknowledgements

This work benefitted from equipment provided by the Microscopy Imaging Center of the University of Bern, and we are grateful for the related support received from Prof. Jens Stein and Dr. Neda Haghayegh. We also thank Luca Molino for assisting in the synthesis and characterization of FXII618 analogues. This work was supported by the Swiss National Science Foundation grants 310030_169526 (to C.H.), 310030_153436 (to A.A.S.), 314730_173127 (to A.A.S.) and 320030_156193 (to R.R.). Support by the NCCR Chemical Biology is also acknowledged. C.L was supported by a Marie Skłodowska-Curie individual fellowship (No 705614).

## Author contributions

J.W., S.J.M, C.L., X.-D.K., and C.H. developed and characterized the FXIIa inhibitor. P.G., J.W., and C.H. cloned and expressed recombinant FXII. J.W., S.J.M., and C.H. designed and performed the pharmacokinetic experiments in rabbits. M.M.A., J.W., R.R., and C.H. designed and performed the pharmacokinetic experiments in pigs. R.P., J.W., X.-D.K., A.A.-S., and C.H. designed and performed the pharmacokinetic, thrombosis, and bleeding experiments in mice. A.C., C.T.D., K.B., K.E.C., N.U., and J.W. designed and performed the artificial lung experiments in rabbits. J.W., C.H., and K.D. wrote the paper. All authors contributed to editing and critical proof-reading of the paper.

## Competing interests

J.W., S.J.M., and C.H. are inventors of a patent covering the FXII900 inhibitor. The remaining authors declare no potential competing interests.
