## [Peer Review File · Nature Communications]

Reviewers' comments:

Reviewer #1 (Remarks to the Author):

Using phage display the authors have developed a peptide inhibitor to FXIIa that excels, existing protein-based inhibitors, in its specificity, selectivity and half-life. Much of the manuscript focuses on improving another inhibitor from the same group (FXII618) to the new inhibitor FXII900. The advantage of FXIIa inhibition in inhibiting thrombosis, without bleeding complications, was further consolidated in mouse models of thrombosis and haemostasis.

1. In Fig. 2, it would have been better to compare FXII900 to FXII901, the latter being the inactive control peptide.
2. In Fig. 2, whereas the concentration-dependent inhibition, in aPTT is very convincing in the rabbit plasma (Fig 2a, bottom left panel), it is less so in human and mouse plasma. Does the inhibitory effect reach saturation, at all, in these two species?
3. Why is it that much higher concentrations of FXII900 are needed to inhibit aPTT in plasma compared to the inhibition against pure FXIIa. This point needs to be addressed, atleast, in the discussion.
4. In Fig. 3, the maximal concentration reached in mouse plasma is around 1 microM (Panel 3a) and aPTT is doubled at this time point. However, this concentration of inhibitor has no effect on aPTT in mouse plasma in Fig. 2a. This discrepancy needs to be addressed.
5. In Fig. 4; what is the effect of the negative control, FXII901, compared to PBS. This is an important control to exclude any spurious effects of the control peptide.
6. Considering that the authors have focussed on improving the specificity of a previously existing peptide it may be useful to do modelling studies to explain why the new peptide is better.

Reviewer #2 (Remarks to the Author):

Inhibition of intrinsic coagulation factor XII (FXII) has the potential to become a safe novel antithrombotic therapy that, unlike the current anti-thrombotic therapies, may not coincide with bleeding as a side-effect. As such, also inhibition of the FXII-downstream coagulation factor XI is of interest and currently under investigation. Antibody and antisense strategies have been developed that successfully inhibit/lower FXII for preclinical testing, however, synthetic small-molecule approaches that are successful (in the preclinic) are not available yet. A small molecule-based therapy may in some respects have advantages over antibody or antisense approaches (regarding synthesis, immunogenicity, and possibly oral availability).

In the current work, Wilbs and colleagues started from their previously generated and reported bicyclic peptide, to generate a novel compound (FXII900) with strong FXIIa inhibition in the sub-nanomolar range. This proved potent for inhibition of human FXIIa, and also for FXIIa from animals allowing preclinical studies (mouse, rabbit, pig). Apart from strong FXIIa inhibition, also high stability of FXII900 was demonstrated using plasmas of the total of four species. The experiments (including compound modifications) that were required to produce the strong inhibitory and high stability compound were well-executed and are clearly and well-described. Convincingly is also demonstrated that FXII900, for the multiple species evaluated, potently impacted intrinsic pathway (prolonged the aPTT), not extrinsic pathway (PT), as anticipated based on specific FXIIa inhibition. Also lack of impact of FXII900 on other related/relevant proteases was convincingly demonstrated.

Pharmacokinetic studies were performed, and for the species analyzed, following subcutaneous/intravenous injection, FXII900 appeared to have a short half-life in the blood circulation. Consequently, a limited time frame is available where FXII900 can impact the functionality of the intrinsic pathway (as analyzed ex vivo using aPTT). The rapid clearance, despite stability of FXII900 in plasma, is attributed to a fast renal clearance.

Subsequently, FXII900 was tested in a preclinical mouse thrombosis model i.e. the ferric chloride model on mesenteric arteries to demonstrate in vivo efficacy. Clearly, FXII900 inhibited thrombosis

in this specific experimental model. Furthermore and importantly, the conditions that proved antithrombotic did not coincided with prolonged bleeding times in the commonly used mouse tail transection bleeding model.

Comments

Abstract:

One important typical characteristic of the FXII900 compound is the short plasma half-life in vivo. This should be addressed in the abstract.

In the abstract is claimed that FXII900 is a promising candidate for safe thromboprotection. In the discussion section, the application of FXII900 is presented with more restriction i.e. applicability in acute procedures such as extracorporeal membrane oxygenation (ECMO) or cardiopulmonary bypass (CPB) surgeries. This should be denoted in the abstract, as the abstract now suggests that FXII900 may also be used in treatment of/prophylaxis in venous and/or arterial thrombosis. However, this would requiring longer half-lives and functional availability of the compound (and optionally also oral bioavailability).

Introduction:

P2. "Mice lacking FXII have reduced risk..." Term " risk" may be not fully appropriate for the experimental mouse condition and could be replaced by wording like: " reduced thrombosis in mouse models of arterial and venous thrombosis"

P3. When summarizing studies on antisense oligonucleotides and antibodies targeting FXII in thrombosis, the type of models used for which the response was recorded could be indicated.

Results:

The results describing the engineering of the compound and testing it's inhibitory potential and stability are well-performed and well-described.

Regarding in vivo administration of FXII900 in mice, rabbits and pigs; what is the effect of FXII900 injection on parameters of acute toxicity? Such as hematology, circulating liver enzymes, plasma inflammation markers, acute phase proteins?

What is known on these class of compounds regarding oral availability? Is FXII900 effective in the circulation of the animals when administered orally? When not tested, or when negative, one can comment on this in the discussion section, since one important reason to go for small synthetic molecule approach, and not for an antibody and antisense approach, is the potential that the compound can orally administered. This would be a major advantage, or even prerequisite, for application in thrombosis prophylaxis.

Given the rather short plasma half-life of FXII900, and consequent restricted duration of impact of FXII900 on the ex vivo analyzed aPTT, FXII900 would be more suitable for applicability in acute procedures such as extracorporeal membrane oxygenation (ECMO) or cardiopulmonary bypass (CPB) surgeries. The authors outlined these applications for FXIIa inhibition already in the introduction and also in the discussion section. Although the antithrombotic effect of FXII900 in the ' ferric chloride model' is strong, this model is rather distant from (any form of) clinical thrombosis and also the suggested/anticipated application. Addition of studies demonstrating the performance of FXII900 in models for which use is anticipated such ECMO and CPB (or stroke) models would definitely strengthen this work.

Figure 4c: the authors could comment on the bi-distribution in the heparin response regarding

blood loss. Moreover, for heparin a mean bleeding time of 30 minutes is observed for all animals included though coinciding for 3 animals with only minimal blood loss (in microliters). Possibly for using the readout 'blood loss' the group size is too low for comparisons. The lines in figure 4c that likely indicate mean values should be clarified in the figure legends. In addition, when having a bi-distribution, one can question whether it is appropriate to use a 'mean' to describe these data and using of parametric testing for comparison.

Discussion:

It is outlined that small synthetic compounds for FXIIa inhibition are attractive. One of the number of advantages of small synthetic compounds may be the limited immunogenicity. On the other hand, the short half-life of FXII900 restricts its anticipated application to use solely in acute situations, with likely single administration, and thus limited chance of resulting in immunoreactivity. Hence, this advantage of the small molecule approach seems to have become irrelevant because of its short plasma half-life and restricted use.

It is mentioned that FXII900 is rapidly cleared by renal filtration. The authors could outline and/or mention the data that support this conclusion.

The half-life of FXII900 could also be discussed in the context of the half-life of other approaches to inhibit FXII and so far developed and published (antisense and antibody 3F7 approach).

Reviewer #3 (Remarks to the Author):

The manuscript by Wilbs et al. describes the optimization of a macrocyclic peptide inhibitor of the coagulation pathway protease factor XII. The authors do a nice job of highlighting the potential for factor XII inhibitors for treatment of a number of conditions in which blood clotting is an issue. The current data from knockout mice and humans deficient for factor XII suggest that this protease will be a possible viable target to block clotting without inducing bleeding.

This work builds on the prior work of the Heinis Lab in which they use a phage display method to identify high affinity binding bicyclic peptides for factor XII. Their prior efforts resulted in the development of FXII618 which was not viable as a lead molecule due to low potency and poor in vivo stability. In the current work, the authors use a combination of rational design and randomization methods to identify a lead molecule with greatly increased potency and stability. The authors show that this new lead molecule, FXII900, appears to be a significant improvement and thus suitable for validation studies in animal models of clotting disorders. The results presented are compelling and show that the molecule has an in vivo half life of 30-90 min in mice, rabbits and pigs. Most importantly, the authors show that the molecule can reduce clot formation in a mouse model without inducing the prolonged bleeding that occurs with heparin sulfate treatment.

Overall, I think this manuscript presents some exciting results for a compound that could be a valuable new clinical lead molecule. The work is all performed with a high degree of rigor and I have no issue with any of the results and how they are presented. The only minor issue is that the molecules still have relatively short half-life in blood, which will limit their utility for some indications. However, this rapid clearance could be a benefit for other situations where only a short-term block of clotting is needed. The study is valuable, although not particularly novel, and is mainly a medicinal chemistry study so I am not sure about the fit of this work for Nature Communications. However, I think the potential impact as a clinical lead molecule may be sufficient to overcome any limitation in novelty of the approach.

I have no specific suggestions and would suggest publication as is.

Point-by-point response to the reviewer comments

Our response: We would like to thank the reviewers for their input and contributions to improve our manuscript. Below is a point-by-point response to their comments and a detailed description of the revisions made in response to their suggestions. We have highlighted all the changes in yellow in the revised manuscript and SI.

Reviewer #1 (Remarks to the Author):

Using phage display the authors have developed a peptide inhibitor to FXIIa that excels, existing protein-based inhibitors, in its specificity, selectivity and half-life. Much of the manuscript focuses on improving another inhibitor from the same group (FXII618) to the new inhibitor FXII900. The advantage of FXIIa inhibition in inhibiting thrombosis, without bleeding complications, was further consolidated in mouse models of thrombosis and haemostasis.

1. In Fig. 2, it would have been better to compare FXII900 to FXII901, the latter being the inactive control peptide.

Our response: In Figure 2a and 2b, we compare aPTT and PT of the new FXII inhibitor FXII900 to its precursor FXII816, in order to show if and to what extent it performs better in inhibiting the intrinsic coagulation pathway. We fully agree with the reviewer that a comparison to FXII901 would be interesting too, as FXII901 is an important negative control. We have performed this additional experiment, measuring aPTT and PT using the same concentrations as for FXII900 shown in Figure 2a and 2b (0.09-30 μM for aPTT, 0.038-120 μM for PT). Pleasingly, FXII901 did not prolong aPTT and PT at the highest concentrations tested. We have added this new data to the Supplementary Figure 8, which contains data about FXII901 (sequence, K_i). We have added the panel "C" below:

2. In Fig. 2, whereas the concentration-dependent inhibition, in aPTT is very convincing in the rabbit plasma (Fig 2a, bottom left panel), it is less so in human and mouse plasma. Does the inhibitory effect reach saturation, at all, in these two species?

Our response: The strong prolongation of aPTT at low FXII inhibitor concentrations and the sharp transition to a maximal effect is a particular effect observed in experiments with rabbit plasma. It is not found for plasmas of other species. The rabbit plasma-specific phenomenon was also reported for antibody-based FXII inhibitors (e.g. reference 14). The molecular basis for this phenomenon is not discussed in any research article and we do not know it. In plasma of other species, aPTT does not reach a plateau at an inhibitor concentration of 30 μ M. We did not test higher concentrations as the inhibitor would start blocking plasma kallikrein, and at mM concentrations most likely also other proteases. We have added the following sentence to make the readers aware of the specific situation observed for aPTT measurements in rabbit plasma:

In rabbit plasma, we observed a strong prolongation of aPTT at low inhibitor concentrations and a sharp transition to a maximal effect. This pattern was previously seen for antibody-based FXIIa inhibitors tested in rabbit plasma¹⁴, and we thus expected that this was a rabbit plasma-specific phenomenon and not based on a particularly strong affinity for rabbit FXIIa.

3. Why is it that much higher concentrations of FXII900 are needed to inhibit aPTT in plasma compared to the inhibition against pure FXIIa. This point needs to be addressed, at least, in the discussion.

Our response: This is a good question. The reason is that 50% of FXII needs to be inhibited to reach the K_i (by definition) while \gg 50% of FXII needs be inhibited in order to reach a 1.5-fold prolongation of aPTT (inhibition of 50% of FXII would most likely not have any effect on aPTT). We estimate that FXII needs to be inhibited $> 99\%$ to prolong aPTT 1.5-fold ($EC_{1.5x}$), and this in turn requires an inhibitor concentration that is > 100 -fold higher than the K_i . We have added the following sentence to describe why we expect much higher values for the K_i than for the aPTT $EC_{1.5x}$:

As expected, the $EC_{1.5x}$ values are much higher than the K_i values because FXII needs to be inhibited nearly fully to prolong aPTT 1.5-fold (we estimate $> 99\%$) while the K_i is by definition the concentration at which 50% of FXII is inhibited.

4. In Fig. 3, the maximal concentration reached in mouse plasma is around 1 microM (Panel 3a) and aPTT is doubled at this time point. However, this concentration of inhibitor has no affect on aPTT in mouse plasma in Fig. 2a. This discrepancy needs to be addressed.

Our response: In the in vitro experiment (Figure 2a), the aPTT of mouse plasma was prolonged 1.26-fold at 0.75 μ M inhibitor and 1.43-fold at 1.5 μ M, respectively (Figure 2a). At a concentration of 1 μ M, we could thus expect a slight prolongation of aPTT in mice, which was the case. As pointed out by the reviewer, the aPTT of plasma taken from inhibitor-injected mice was slightly higher than expected. However, we think that the difference is within the error range of inhibitor concentration determination by LC-MS in plasma/aPTT measurement. We are confident that the observed effects in mice are resulting from the inhibitor, as there is a nice correlation between the

inhibitor concentration in vivo and aPTT, and a peak for both parameters around the same time period (Figure 3a).

5. In Fig. 4; what is the effect of the negative control, FXII901, compared to PBS. This is an important control to exclude any spurious effects of the control peptide.

Our response: We fully agree with the reviewer that it is important to ensure that the negative control peptide FXII901 ($K_i = 39 \pm 14 \mu\text{M}$, > 100,000-fold worse than FXII900) has no effect on blood coagulation. As described above in the answer to question 1, we have now tested the effect of FXII901 on aPTT and PT and found that it is inactive (please see Supplementary Fig. 8c).

For the ferric chloride thrombosis study, we have used the inactive peptide FXII901 as negative control and we did not use PBS as a second negative control. In experiments that we had performed at the beginning of the study to establish the mouse model in our labs, we injected nothing and those mice showed a similar extent of coagulation (time of coagulation, % of mice forming a thrombus) as mice that were treated with the inactive peptide FXII901. We therefore think that the FXII901 peptide has no effect on thrombus formation in the ferric chloride mouse model. In addition, the time and extent of thrombosis observed for FXII901-treated mice was comparable to that of other studies in which vehicle-treated mice were exposed to the same % of FeCl_3 (7.5%; e.g. Wang, X. et al., *Journal of Thrombosis and Haemostasis*, 2004). We now provide this information in the manuscript:

The time point and extent of clot formation observed for control peptide were as expected based on previous studies in which vessels of non-treated mice were exposed to 7.5% FeCl_3 ⁴¹.

For the mouse tail-bleeding study (results shown in Figure 4c), we have now tested, in addition to the effect of PBS, also the effect of the negative control peptide FXII901, as suggested by the reviewer. This experiment showed comparable bleeding times and blood volume losses for mice treated with PBS, FXII900 and FXII901. The new Figure 4c is the following one:

6. Considering that the authors have focussed on improving the specificity of a previously existing peptide it may be useful to do modelling studies to explain why the new peptide is better.

Our response: We have done such a modeling study for the precursor of FXII900, the inhibitor FXII618, as reported in Baeriswyl, V. et al., *ACS Chemical Biology*, 2013. Given the high structural similarity of the peptides, we assumed that FXII900 occupies the same space and forms the same

key interactions as FXII618. A limitation with the model is that we could determine with high certainty only the binding mode of the amino acids Phe3 to Pro6. Unfortunately, the model does not provide any information about the amino acid positions that were improved to obtain FXII900. We now refer to this structural model and its limitations as follows:

We had previously developed a structural model of the precursor peptide FXII618 bound to FXIIa, in which we could determine with high confidence the positions and interactions of the amino acids Phe3 to Pro6³³. The newly identified beneficial amino acid substitutions lie outside this region and we could thus not use the model to rationalize the molecular basis of the achieved affinity enhancements.

Reviewer #2 (Remarks to the Author):

Inhibition of intrinsic coagulation factor XII (FXII) has the potential to become a safe novel antithrombotic therapy that, unlike the current anti-thrombotic therapies, may not coincide with bleeding as a side-effect. As such, also inhibition of the FXII-downstream coagulation factor XI is of interest and currently under investigation. Antibody and antisense strategies have been developed that successfully inhibit/lower FXII for preclinical testing, however, synthetic small-molecule approaches that are successful (in the preclinic) are not available yet. A small molecule-based therapy may in some respects have advantages over antibody or antisense approaches (regarding synthesis, immunogenicity, and possibly oral availability).

In the current work, Wilbs and colleagues started from their previously generated and reported bicyclic peptide, to generate a novel compound (FXII900) with strong FXIIa inhibition in the sub-nanomolar range. This proved potent for inhibition of human FXIIa, and also for FXIIa from animals allowing preclinical studies (mouse, rabbit, pig). Apart from strong FXIIa inhibition, also high stability of FXII900 was demonstrated using plasmas of the total of four species. The experiments (including compound modifications) that were required to produce the strong inhibitory and high stability compound were well-executed and are clearly and well-described. Convincingly is also demonstrated that FXII900, for the multiple species evaluated, potentially impacted intrinsic pathway (prolonged the aPTT), not extrinsic pathway (PT), as anticipated based on specific FXIIa inhibition. Also lack of impact of FXII900 on other related/relevant proteases was convincingly demonstrated.

Pharmacokinetic studies were performed, and for the species analyzed, following subcutaneous/intravenous injection, FXII900 appeared to have a short half-life in the blood circulation. Consequently, a limited time frame is available where FXII900 can impact the functionality of the intrinsic pathway (as analyzed ex vivo using aPTT). The rapid clearance, despite stability of FXII900 in plasma, is attributed to a fast renal clearance.

Subsequently, FXII900 was tested in a preclinical mouse thrombosis model i.e. the ferric chloride model on mesenteric arteries to demonstrate in vivo efficacy. Clearly, FXII900 inhibited thrombosis in this specific experimental model. Furthermore and importantly, the conditions that proved antithrombotic did not coincide with prolonged bleeding times in the commonly used mouse tail transection bleeding model.

Comments

Abstract:

One important typical characteristic of the FXII900 compound is the short plasma half-life in vivo. This should be addressed in the abstract.

In the abstract is claimed that FXII900 is a promising candidate for safe thromboprotection. In the discussion section, the application of FXII900 is presented with more restriction i.e. applicability in acute procedures such as extracorporeal membrane oxygenation (ECMO) or cardiopulmonary bypass (CPB) surgeries. This should be denoted in the abstract, as the abstract now suggests that FXII900 may also be used in treatment of/prophylaxis in venous and/or arterial thrombosis. However, this would requiring longer half-lives and functional availability of the compound (and optionally also oral bioavailability).

Our response: We agree with the reviewer. We have changed the abstract to be more specific about the potential applications of the FXII inhibitor.

... and that the inhibitor FXII900 is a promising candidate for safe thromboprotection in acute medical conditions.

Introduction:

P2. "Mice lacking FXII have reduced risk..." Term "risk" may be not fully appropriate for the experimental mouse condition and could be replaced by wording like: "reduced thrombosis in mouse models of arterial and venous thrombosis"

Our response: We fully agree. We have changed the wording:

Mice lacking FXII display reduced thrombosis in mouse models of injury-induced arterial and venous thrombosis...

P3. When summarizing studies on antisense oligonucleotides and antibodies targeting FXII in thrombosis, the type of models used for which the response was recorded could be indicated.

Our response: We have added this information:

Concordantly, the reduction of FXII expression by antisense oligonucleotides suppressed thrombosis in arterial and venous thrombosis mouse models and catheter thrombosis in rabbits^{12,13}. The inhibition of FXII by protein-based inhibitors, such as antibodies^{14,15} or insect- and plant-derived proteins¹⁶⁻¹⁸, also reduced thrombosis in mouse, rat, rabbit and primate models of induced arterial or venous thrombosis and showed potential avenues for therapeutic anticoagulation.

Results:

The results describing the engineering of the compound and testing its inhibitory potential and stability are well-performed and well-described.

Regarding in vivo administration of FXII900 in mice, rabbits and pigs; what is the effect of FXII900 injection on parameters of acute toxicity? Such as hematology, circulating liver enzymes, plasma inflammation markers, acute phase proteins?

Our response: We have not observed any adverse effects in any experiment with the three species. Best controlled in terms of side effects were the PK studies in which the animals were not anesthetized. We have now included a statement in which we write that no adverse effects were observed in any of the three species:

In all the studies performed with the three species, we did not observe any signs of toxicity or other adverse effects.

What is known on these class of compounds regarding oral availability? Is FXII900 effective in the circulation of the animals when administered orally? When not tested, or when negative, one can comment on this in the discussion section, since one important reason to go for small synthetic molecule approach, and not for an antibody and antisense approach, is the potential that the compound can orally administered. This would be a major advantage, or even prerequisite, for application in thrombosis prophylaxis.

Our response: We have not tested the oral availability of FXII900. While the inhibitor has a high stability and might survive to some extent the high proteolytic pressure in the gastrointestinal tract, the inhibitor would most likely not cross the epithelial layer and thus not reach the blood stream. We have added the following statement in the discussion:

Given the high proteolytic stability, FXII900 could potentially resist to some extent proteases of the stomach and intestines, but due to the multiple charges, it would unlikely be able to cross the gastrointestinal epithelium, preventing its oral administration.

Given the rather short plasma half-life of FXII900, and consequent restricted duration of impact of FXII900 on the ex vivo analyzed aPTT, FXII900 would be more suitable for applicability in acute procedures such as extracorporeal membrane oxygenation (ECMO) or cardiopulmonary bypass (CPB) surgeries. The authors outlined these applications for FXIIa inhibition already in the introduction and also in the discussion section. Although the antithrombotic effect of FXII900 in the 'ferric chloride model' is strong, this model is rather distant from (any form of) clinical thrombosis and also the suggested/anticipated application. Addition of studies demonstrating the performance of FXII900 in models for which use is anticipated such ECMO and CPB (or stroke) models would definitely strengthen this work.

Our response: We fully agree with this analysis of the reviewer on the application range of the inhibitor. Given the half-life and application route (i.v.), ECMO and CPB are definitively the most attractive applications of the inhibitor in its current form. We had already tested FXII900 in a rabbit ECMO model (the data was provided in the submission as a PDF document) and this study showed that infusion of the inhibitor over four hours in rabbits can efficiently suppress contact

activation, as measured by aPTT. ACT and blood clotting. Our collaborator in this study, Prof. Keith Cook and his team have agreed to include the data in this manuscript. We have added the following authors to the author list:

Alida Cooke⁴, Caitlin T. Demarest⁴, Kalliope Roberts⁴, Nao Umei⁴

⁴Department of Biomedical Engineering, Carnegie Mellon University, Pittsburgh, Pennsylvania

The additional manuscript chapter reads as follows:

FXII900 provides bleeding-free anticoagulation in artificial lungs

To test the clinical potential of FXII900 for inhibiting FXII-driven coagulation in ECMO, we tested the inhibitor in an artificial lung rabbit model. New Zealand White rabbits were anesthetized, either treated intravenously with FXII900 (n = 4, 2 mg/kg bolus and 0.075 mg/kg/min infusion for four hours), left untreated (n = 3), or treated with heparin for comparison (n = 2, infusion of 60 IU/hr and rate adjustments to maintain ACT within a clinical range of 220-300 s), and were mounted for a four-hour period to a veno-venous ECMO configuration using a polymethylpentene (PMP) fiber artificial lung system. ACT and aPTT were measured over the entire duration of the experiment to test if infusion of the inhibitor can prolong these coagulation parameters for several hours, which would be a requirement in applications such as ECMO. The resistance through the lung device system, a measure for occlusive thrombus formation, and clot formation in the lung device analyzed after four hours, were measured too (Fig. 5; data for the individual rabbits is shown in Supplementary Fig. 12).

Treatment with FXII900 prolonged the coagulation parameters aPTT and activated clotting time (ACT) around 10-fold during the entire course of the experiment. This was to a much larger extent than those for heparin, which were prolonged 1.5 and 2-fold, the range recommended for heparin anticoagulation (Fig. 5a and 5b). For most aPTT measurements using FXII900, no coagulation was observed at 240 seconds, the maximal aPTT value measured with this device (Supplementary Fig. 12a). In untreated rabbits, the coagulation times remained at baseline values of around 20 seconds (aPTT) and 170 seconds (ACT) throughout the experiment. Because increased pressure at the inlet of the artificial lung indicates clogging that can be caused by blood coagulation, we measured the resistance of the artificial lung, indicated by the difference in pressure at the inlet and outlet of the lung divided by the flow. In all rabbits treated with FXII900, the resistance remained at the baseline of 50 mmHg/L/min during the entire experiment, except for one rabbit in which the resistance started to increase strongly after two hours at baseline (Fig. 5c and Supplementary Fig. 12c). This was in stark contrast to the untreated animals in which the resistance increased in the first hour and doubled for all rabbits from 50 mmHg/L/min to 100 mmHg/L/min during the course of the experiment.

Because excessive bleeding is a major concern with standard anticoagulation in ECMO systems, we analyzed if FXII900 affected the hematological parameters of bleeding time and platelet count. Rabbits treated with the inhibitor showed completely normal bleeding times throughout the experiment (4–5 min; Fig. 5c and Supplementary Fig. 12d). In contrast, the group treated with heparin at a clinically relevant dose showed 2-fold prolonged bleeding times at all the time points measured (5-10 min; Fig. 5d and Supplementary Fig. 12d). The platelet count was essentially the same in the three groups, decreasing around two-fold from an initial level of 2×10^8 cells/ml

(before connecting the lung) and remaining constant over the entire course of the experiment (Fig. 5e and Supplementary Fig. 12e). Analysis of the artificial lungs at the end of the experiment showed that the inhibitor- and heparin-treated rabbits had a reduced amount of clotted blood. The volume of the clot in relation to the device volume was $10 \pm 6\%$ for the FXII900-treated rabbits, which was significantly smaller than that of untreated rabbits ($37 \pm 10\%$; $p = 0.03$), but it was slightly larger than that of the heparin-treated rabbits (5% ; Fig 5f and Supplementary Fig. 12f).

Fig. 5. Bleeding-free anticoagulation in an artificial lung model in rabbits. Rabbits were connected veno-venous to an artificial lung system for four hours. FXII900 was injected as a bolus (2 mg/kg) before the start of the extracorporeal circulation and as a constant infusion (0.075 mg/kg/min) over the full time course of the experiment ($n = 4$; data shown in blue). Rabbits in the control group were untreated ($n = 3$, data shown in red). Rabbits in the heparin group ($n = 2$) were initially infused with one IU/min heparin and the drip rate was adjusted to maintain ACT values between 220 and 300 s. Values indicated for the time point 0 refer to measurements made before connecting the artificial lung. Mean \pm SD is indicated for the four rabbits in the FXII900-treated group and the three rabbits in the control group. Mean values are indicated for the two rabbits in the heparin group. Data from individual rabbits is shown in Supplementary Fig. 12. **a**, aPTT. Coagulation times of 240 s indicate that plasma did not coagulate at this time. **b**, ACT. Clotting times of 1500 s indicate that blood did not coagulate at this time. **c**, Resistance calculated based on pressure at the inlet and outlet of the device and the flow rate. **d**, Bleeding time measured by incision-provoked injuries at the ears. **e**, Platelet count. **f**, Volume of blood clots indicated in % of volume of the artificial lungs.

Given the addition of these additional results, we have also added the following paragraph in the discussion section:

Finally, we further tested if FXII900 could suppress the coagulation activated by the membrane of artificial lungs using a rabbit model. Heart-lung machines are used for heart surgery because of the difficulty of operating on the beating heart such that approximately half a million cardiac operations performed per year in the US use CPB. Similar, more compact systems are also used to support patients with cardiac and respiratory dysfunction, with more than 7,000 patients treated annually with extracorporeal life support.⁴² The contact of blood with the hydrophobic polymer surfaces found in these machines induces FXII contact activation and thus coagulation. This coagulation is currently suppressed using high doses of heparin, which bears bleeding risks. An analysis of nearly 80,000 ECMO patients showed that hemorrhage is the major adverse event in these situations, with bleeding from surgical sites occurring in 6.3% to 29.3% of patients depending on the patient group.⁴² Moreover, both device thrombosis and patient bleeding lead to significant increases in patient mortality.⁴³ We found that continuous infusion of FXII900 efficiently prolonged the coagulation parameters ACT and aPTT over four hours and thus the entire course of the experiment. The inhibitor reduced coagulation in an artificial lung as measured by the resistance of the device, which remained low and indicated reduced clogging by coagulation and a reduction in the total volume of the blood clots in the device, which was more than three-fold lower in treated rabbits. Importantly, all rabbits treated with FXII900 showed normal bleeding times. These findings indicated that a small molecule FXIIa inhibitor is suitable for reducing blood coagulation in CPB. While heparin may still be required to suppress coagulation triggered through routes other than FXII, the application of a small molecule FXIIa inhibitor may allow for a reduced heparin dose, which would reduce the bleeding risks. In addition to its role in thrombosis in ECMO and CPB, FXIIa initiates the inflammatory kallikrein-kinin system.¹⁹ Bradykinin plasma concentrations are largely elevated in CPB and ECMO patients, so targeting FXIIa may provide an additional anti-inflammatory benefit to these patients, to reduce or prevent adverse effects, such as organ damage.

The materials and methods part of the additional experiment is the following one:

Artificial lungs

Mini-lungs were created that allowed for blood exposure to the fibers in the artificial lung, but for construction simplicity, do not transfer gas. Polymethylpentene (PMP) fiber (3M) with 50% porosity, a 380 μ m outer diameter, and two layers at 30° cross angles were cut into 1.78 \times 1.78 cm squares. Five of these layers were put together, making sure the fibers all ran in the same direction, and they were sealed together into a fiber bundle using a hot plate on each side of the square sheets. These fiber bundles were melted such that they had a 1.57 \times 1.57 cm square frontal area perpendicular to the flow. Eight 5-layer "chiclets" were placed together to make up the fiber bundle of one mini lung device, giving a final surface area of 263 cm². These fiber bundles were placed in a square plastic housing of 3.05 cm in length, making sure that the fiber bundles fit tightly in the device to prevent shunting around the fiber bundle. This housing was attached to plastic end caps that were placed on each end with a 1/8"-barbed tube fitting for 3/16"-ID tubing to allow the device to be connected to tubing in the circuit. These end connectors were coated with Teflon tape before connecting them to the plastic end caps to prevent leakage. The device

was held together with two screws going from end cap to end cap. The entire device was secured with silicone to eliminate leakage. The silicone was left to dry for 24 hours, and the device leak tested as follows. Filtered deionized water was run through the device to check for leaks within the device. If no leaks were found, the device was left to dry with filtered air running through the device for 24 hours.

Extracorporeal circulation in rabbits

New Zealand white rabbits of 3.2–4.2 kg (Charles River) were anesthetized via intramuscular injections of ketamine (30 mg/kg) and xylazine (5 mg/kg). One of the ear veins was catheterized via a 24G winged catheter, and the rabbits were intubated with a 3.0 endotracheal tube. The animals were kept anesthetized throughout the four-hour experiment via inhaled isoflurane (2%) and were ventilated with a peak inspiratory pressure (PIP) of < 20 cmH₂O, positive end expiratory pressure of 5 mmHg, tidal volume of 4–6 ml/kg, and a respiratory rate of 22–60 breaths/min. Tidal volume and respiratory rate were adjusted to maintain normal arterial blood gases and the listed PIP. Phenylephrine was applied intravenously at a rate of 0.5–5 µg/kg min to maintain blood pressure. For monitoring blood pressure and collecting blood samples, the rabbits' right or left carotid arteries were cannulated using a 16-gauge angiocatheter (Becton Dickinson) and were secured with silk ties. The device and circuit were first primed with filtered CO₂ and then with saline (NaCl, 0.9% [w/v]) containing solumedrol (30 mg/kg). At this point, rabbits of the inhibitor-treated group were injected with 2 mg/kg of FXII900 as a 2 mM solution in PBS (pH 7.4) via the ear vein and the circuit was connected to the rabbits via a venous/venous ECMO configuration using a 14-gauge angiocatheter (Becton Dickinson) and a 6" pressure tubing that was cut to length from a 24" pressure tubing (Edwards Lifesciences) in the right and left internal jugular veins, respectively. The circuit was placed in a roller pump (Cobe), and the blood flow was set to 45 ml/min. After connecting the extracorporeal circuit, rabbits of the inhibitor-treated group were infused with 0.075 mg/kg/min FXII900 via the ear vein for the entire duration of the study. Rabbits treated with heparin were initially infused with 60 IU/hr (100 IU/ml) starting before connection to the circuit. The drip rate was adjusted based on the following nomogram to maintain the ACT in a range between 220 and 300 s. ACT > 320 s: decrease infusion rate by 12 IU/hr, 320s > ACT > 300 s: decrease infusion rate by 6 IU/hr, 300 s > ACT > 220 s: maintain heparin infusion rate, 220s > ACT > 200s: increase infusion rate by 6 IU/hr, 200s > ACT > 180s: increase infusion rate by 12 IU/hr, ACT < 180s bolus 100 U (1ml). After four hours, the animals were euthanized by potassium chloride (2 mg/kg, IV). At the end of the experiment, 5 ml of heparin was run through the artificial lung while it was still connected to the rabbits. Then the circuit was removed, and the device was washed with saline carefully so that no clot shedding occurred. Saline was run through the device until the drained portion the effluent was clear. Clot volume was measured by measuring the volume of the device prior to the experiment and again at the end of the experiment after washing. This was done by completely filling the device with saline and recording this amount of saline as the device volume. The difference between the beginning and ending volume was determined to be the clot volume. From the clot volume, the percent of clot within the device was calculated.

Data acquisition during extracorporeal circulation

Platelet and white blood cell counts, hematocrit, arterial blood gases (ABG), ACT, aPTT, fibrinopeptide A (FPA), device resistance, and bleeding time were measured prior to circuit attachment and at 10, 30, 120, and 240 minutes following the initiation of ECMO. For the platelet counts, a syringe with 0.05 ml of 3.2% sodium citrate (w/v) was used to draw 0.45 ml of blood for a total volume of 0.5 ml. This was then centrifuged at 60 g for 10 minutes, and 20 μ l of the plasma was placed in 20 ml of ISOTON® diluent and counted using a Coulter Counter (Beckman Coulter, Inc. Brea, CA) with a 50 μ l aperture tube. For counting, cells were considered platelets if their diameters were 1.8–5.6 μ m. For white blood cell counts, 40 μ l of whole blood was placed in 20 ml of the ISOTON® diluent. Six drops of ZAP-OGLOBIN lysing solution were added to the mixture and mixed gently. This was allowed to sit for two minutes. The white blood cells were also counted using the Coulter Counter (Beckman Coulter) with a 50 μ l aperture tube where any particle above 3.6 μ m in diameter was considered a white blood cell. Arterial blood gases were measured by drawing 0.4 ml of blood into a heparinized syringe and run using an arterial blood gas analyzer (ABL800 FLEX, Radiometer). The ACT and hematocrit were measured by collecting 0.5 ml of blood. The ACT was measured using a Hemochron analyzer with tubes containing glass beads as the activator. The hematocrit was measured via capillary centrifugation. For aPTT measurements, blood samples (1.8 ml) were collected at different time points into tubes containing sodium citrate (0.2 ml, 3.2% [w/v] sodium citrate) and immediately processed to plasma by centrifugation at 2000 g for 15 minutes at 4°C. The samples were then analyzed as described above. All platelet counts, WBC counts, FPA, and FXIIa levels were corrected for hemodilution by adjusting the raw values based on the hematocrit. The inlet and outlet pressure and blood flow rate were measured using a Biopac system (Aero Camino Goleta, CA) and pressure transducers at the inlet and outlet of the device (Edwards Lifesciences), and the resistance was calculated with the standard $R = (P_i - P_o)/Q$ where P_i is the inlet pressure in mmHg, P_o is the outlet pressure in mmHg, and Q is the flow in L/min. The bleeding time was measured by cutting small incisions of 4–5 mm at different sites of the right or left ear in each animal. Blood from the incision was removed with gauze every 30 seconds, and the time until the bleeding stopped was measured.

Figure 4c: the authors could comment on the bi-distribution in the heparin response regarding blood loss. Moreover, for heparin a mean bleeding time of 30 minutes is observed for all animals included though coinciding for 3 animals with only minimal blood loss (in microliters). Possibly for using the readout 'blood loss' the group size is too low for comparisons.

Our response: We did not expect to find the heterogeneous result (bi-distribution) for the blood loss in the heparin control of the mouse tail-bleeding experiment. We have now repeated the control experiment with heparin using four additional mice, which all showed medium to high blood loss. We have added the data of the additional four mice to the right graph in Figure 4c. Given that technically nothing went wrong with the six mice of the first experiment, we decided to not replace the data but to add the new data.

We agree with the reviewer that the number of mice may be too small to compare the blood loss of mice treated with heparin with that of mice treated with PBS, FXII901, or inhibitor. We have changed the text as follows:

Mice injected with heparin bled essentially continuously throughout the monitored 30 min period, which is similar to previous reports⁴¹ and showed either a small (3 mice) or large loss of blood (3 mice). Given the heterogeneous result for the blood loss, we performed the heparin control with four additional mice, which showed medium to high blood loss.

Given the suggestion of reviewer 1 to assess the effect of the neg. control peptide FXII901 in the tail-bleeding assay, we performed this additional experiment and added the data to Figure 4c, as shown in the following figure:

The lines in figure 4c that likely indicate mean values should be clarified in the figure legends. In addition, when having a bi-distribution, one can question whether it is appropriate to use a 'mean' to describe these data and using of parametric testing for comparison.

Our response: We fully agree with this comment. We have changed the right panel of Figure 4c by removing the mean value and standard deviation of the heparin control, and explain in detail the statistical analysis in the figure legend:

Mean values and standard deviations are indicated. *** $p < 0.001$, n.s., not significant. Mice treated with heparin showed either a small or medium-to-large loss of blood, and a mean value is thus not indicated.

Discussion:

It is outlined that small synthetic compounds for FXIIa inhibition are attractive. One of the number of advantages of small synthetic compounds may be the limited immunogenicity. On the other hand, the short half-life of FXII900 restricts its anticipated application to use solely in acute situations, with likely single administration, and thus limited chance of resulting in immunoreactivity. Hence, this advantage of the small molecule approach seems to have become irrelevant because of its short plasma half-life and restricted use.

Our response: We agree. We therefore do not mention this potential advantage any longer and have changed the text accordingly:

The synthetic nature and the small size ~~promise a minimal risk of immunogenicity and~~ allows for efficient production of the compound.

It is mentioned that FXII900 is rapidly cleared by renal filtration. The authors could outline and/or mention the data that support this conclusion.

Our response: We have studied the clearance of several bicyclic peptides before and we found always renal clearance. Given the short half-life, we assumed that FXII900 is cleared by the kidney too, but we have actually not experimentally tested this. We have changed the text to make clear that we have not experimentally analyzed the route of clearance:

FXII900 is rapidly cleared upon intravenous administration, most likely by renal filtration, as expected based on the small size and the polar structure.

The half-life of FXII900 could also be discussed in the context of the half-life of other approaches to inhibit FXII and so far developed and published (antisense and antibody 3F7 approach).

Our response: We agree. We have added the following discussion:

For controlling the activity of FXII over several days, weeks or months, FXII900 is not suited and other therapeutic modalities such as antibody-based FXII inhibitors or antisense oligonucleotides are clearly better options.

Reviewer #3 (Remarks to the Author):

The manuscript by Wilbs et al. describes the optimization of a macrocyclic peptide inhibitor of the coagulation pathway protease factor XII. The authors do a nice job of highlighting the potential for factor XII inhibitors for treatment of a number of conditions in which blood clotting is an issue. The current data from knockout mice and humans deficient for factor XII suggest that this protease will be a possible viable target to block clotting without inducing bleeding.

This work builds on the prior work of the Heinis Lab in which they use a phage display method to identify high affinity binding bicyclic peptides for factor XII. Their prior efforts resulted in the development of FXII618 which was not viable as a lead molecule due to low potency and poor in vivo stability. In the current work, the authors use a combination of rational design and randomization methods to identify a lead molecule with greatly increased potency and stability. The authors show that this new lead molecule, FXII900, appears to be a significant improvement and thus suitable for validation studies in animal models of clotting disorders. The results presented are compelling and show that the molecule has an in vivo half life of 30-90 min in mice, rabbits and pigs. Most importantly, the authors show that the molecule can reduce clot formation in a mouse model without inducing the prolonged bleeding that occurs with heparin sulfate treatment.

Overall, I think this manuscript presents some exciting results for a compound that could be a valuable new clinical lead molecule. The work is all performed with a high degree of rigor and I have no issue with any of the results and how they are presented. The only minor issue is that

the molecules still have relatively short half-life in blood, which will limit their utility for some indications. However, this rapid clearance could be a benefit for other situations where only a short-term block of clotting is needed. The study is valuable, although not particularly novel, and is mainly a medicinal chemistry study so I am not sure about the fit of this work for Nature Communications. However, I think the potential impact as a clinical lead molecule may be sufficient to overcome any limitation in novelty of the approach.

I have no specific suggestions and would suggest publication as is.

REVIEWERS' COMMENTS:

Reviewer #1 (Remarks to the Author):

None

Reviewer #2 (Remarks to the Author):

All issues raised were very well-addressed. Additional data are provided that further strengthen the work. No further comments